# Interaction-Breaking Adversarial Learning Framework for Robust Multi-Agent Reinforcement Learning

**Sunwoo Lee** [1]   **Mingu Kang** [1]   **Yonghyeon Jo** [1]   **Seungyul Han** [1*]

## Abstract

Cooperation is central to multi-agent reinforcement learning (MARL), yet learned coordination can be fragile when external perturbations disrupt inter-agent interactions. Prior robust MARL methods have primarily considered value-oriented attacks, leaving a gap in robustness when interaction structures themselves are corrupted. In this paper, we propose an interaction-breaking adversarial learning (IBAL) framework that takes an information-theoretic view to construct attacks that impede coordination by perturbing agents' observations and actions, and trains agents to perform reliably under such disruptions. Empirically, our approach improves robustness over existing robust MARL baselines across diverse attack settings and yields stronger performance even under agent-missing scenarios. Our code is available at https://sunwoolee0504.github.io/IBAL.

## 1. Introduction

Cooperative multi-agent reinforcement learning (MARL) studies sequential decision making with multiple agents that coordinate to optimize a shared objective, with applications such as competitive games, multi-robot control, and resource management (Yun et al., 2022; Chen et al., 2023; Orr & Dutta, 2023; Jendoubi & Bouffard, 2023). Since each agent typically observes only a partial view of the environment, the centralized training and decentralized execution (CTDE) framework (Oliehoek et al., 2008) is widely adopted, learning a centralized value function during training while deploying decentralized policies conditioned on local observations. A central challenge in CTDE is credit assignment, and representative approaches include Value Decomposition Networks (VDN) (Sunehag et al., 2017),

QMIX (Rashid et al., 2020b), and related extensions (Wang et al., 2020a; Jo et al., 2026), which aim to align individual decisions with optimal joint behavior through joint action-value learning. Despite their success, CTDE-based methods can suffer substantial performance degradation under environmental perturbations or adversarial attacks (Moos et al., 2022; Guo et al., 2022), motivating robustness as a central requirement in cooperative MARL.

A growing body of work has investigated robustness in MARL (Lin et al., 2020; Li et al., 2019; He et al., 2023). Existing approaches typically pursue robustness by modeling uncertainty and perturbations during policy learning (Zhang et al., 2021b; Goodfellow et al., 2014), or by considering adversarial manipulations that degrade agent behavior, for example by biasing agents toward suboptimal actions (Bukharin et al., 2023; Yuan et al., 2023; Lee et al., 2025) or by corrupting communication channels (Xue et al., 2021) used for coordination. While effective under their targeted perturbation models, these methods often fail to capture breakdowns in the interaction structure underlying coordination. As a result, robustness against interaction-level failures can remain limited when agents cannot reliably interact or when coordinated attacks disrupt their dependencies, causing substantial performance degradation in tightly coupled cooperative tasks.

Such attacks are particularly critical under CTDE, where effective decision making hinges on coordination rather than independent behaviors. To address this issue, we propose **Interaction-Breaking Adversarial Learning (IBAL)**, a robust MARL framework that characterizes inter-agent interaction and trains policies to remain effective under interaction-breaking attacks. Specifically, we partition agents into two groups and use mutual information (MI) to quantify cross-group influence, yielding an attacker that occludes cross-group observations and perturbs actions to hinder coordination. IBAL enables policies to learn and act reliably even when agents cannot observe each other or when coordination partially collapses. Empirically, IBAL improves robustness over prior robust MARL methods across diverse perturbations and attacks, and achieves substantially stronger performance under non-parametric perturbations where some agents are entirely absent.

---

[1]Graduate School of Artificial Intelligence, UNIST, Ulsan, South Korea. Correspondence to: Seungyul Han <syhan@unist.ac.kr>.

*Proceedings of the 43rd International Conference on Machine Learning*, Seoul, South Korea. PMLR 306, 2026. Copyright 2026 by the author(s).

Our contributions are summarized as follows:

- **Interaction-Breaking Attack:** We introduce a MI-based adversarial attack that suppresses cross-group influence by jointly occluding cross-group observations and perturbing coordinated actions.

- **POMDP Formulation and Adversarial Learning:** We define a new POMDP under the proposed attack and show that it is equivalent, in terms of the value function, to the induced Dec-POMDP with perturbed dynamics, enabling robust policy learning.

- **Empirical Evaluation and Analyses:** We show that IBAL outperforms prior methods under diverse attacks and perturbations, including agent-missing settings, and analyze when and why the proposed attack is effective.

## 2. Related Works

**Robust Multi-Agent RL.** Robust MARL addresses perturbations in multi-agent environments. One common direction improves robustness through regularization-based objectives (Lin et al., 2020; Li et al., 2023b; Wang et al., 2023; Bukharin et al., 2023; Li et al., 2025) and distributional RL methods that explicitly model uncertainty (Li et al., 2020; Xu et al., 2021; Du et al., 2024; Geng et al., 2024). Another line of work revisits equilibrium concepts, developing robust variants of Nash equilibria tailored to multi-agent systems (Zhang et al., 2020b; Li et al., 2023a). In addition, max-min robust optimization (Chinchuluun et al., 2008; Han & Sung, 2021) has been integrated into MARL formulations to learn policies resilient to worst-case perturbations (Li et al., 2019; Wang et al., 2022).

**Adversarial Learning Frameworks.** A substantial line of research in RL improves robustness (Nilim & El Ghaoui, 2005; Moos et al., 2022; Lee et al., 2026) by training agents to withstand adversarial perturbations. In single-agent MDPs, attacks commonly target states or observations (Pattanaik et al., 2017; Zhang et al., 2020a; 2021a; Qiaoben et al., 2024), actions (Pinto et al., 2017; Tessler et al., 2019; Tan et al., 2020; Lee et al., 2021; Liu et al., 2024), rewards (Bouhaddi & Adi, 2023; Rakhsha et al., 2021; Xu et al., 2024), or environment dynamics (Chae et al., 2022). These threat models extend to multi-agent settings via state or observation uncertainties (Han et al., 2022; He et al., 2023; Zhou et al., 2025), action attacks (Yuan et al., 2023), and reward perturbations (Kardeş et al., 2011). Prior work also studies adversarial effects in value-decomposition frameworks (Phan et al., 2021), attacks on critical agents (Yuan et al., 2023; Zhou et al., 2024b; Lee et al., 2025), and vulnerabilities in inter-agent communication (Xue et al., 2021; Tu et al., 2021; Sun et al., 2023). Adversarially shaped frameworks have also been used to improve zero-shot human-AI coordination (Yan et al., 2023; Kang et al., 2026).

**Information-Theoretic Approaches for MARL.** Mutual information has been widely used in MARL to estimate and exploit inter-agent influence (Jaques et al., 2019; Li et al., 2022; Ye & Lu, 2023; Zhou et al., 2024a; Park et al., 2026). Building on this idea, prior work leverages MI for structured exploration and role diversity under parameter sharing (Mahajan et al., 2019; Li et al., 2021; Jo et al., 2024), and uses influence-based signals to guide and stabilize inter-agent communication (Wang et al., 2020b; Guan et al., 2022; Ding et al., 2024; Bae et al., 2026). In contrast, we use MI to quantify cross-group influence and design an attacker that disrupts coordination by minimizing this influence, rather than targeting values.

## 3. Background

### 3.1. Dec-POMDP and Value-based CTDE Setup

A fully cooperative multi-agent task is modeled as a decentralized partially observable Markov decision process (Dec-POMDP) (Oliehoek et al., 2016), $\mathcal{M} = \langle \mathcal{N}, \mathcal{S}, \mathcal{A}, P, \Omega, O, R, \gamma \rangle$, where $\mathcal{N} = \{1, \ldots, n\}$ is the set of agents, $\mathcal{S}$ the global state space, $\mathcal{A} = \prod_{i=1}^{n} \mathcal{A}^i$ the joint action space, and $P$ the transition probability. At time $t$, agent $i$ receives an observation $o_t^i = O(s_t, i) \in \Omega$ and selects an action $a_t^i \in \mathcal{A}^i$ according to a decentralized policy $\pi^i(\cdot \mid \tau_t^i)$, where $\tau_t^i$ is the local trajectory. The joint action $\boldsymbol{a}_t = \langle a_t^1, \ldots, a_t^n \rangle$ is sampled from the joint policy $\boldsymbol{\pi} := \prod_{i=1}^{n} \pi^i$, leading to a transition $s_{t+1} \sim P(\cdot \mid s_t, \boldsymbol{a}_t)$ and a shared reward $r_t := R(s_t, \boldsymbol{a}_t, s_{t+1})$. The objective is to maximize the discounted return $\sum_{t=0}^{\infty} \gamma^t r_t$. We adopt CTDE framework, in which a centralized critic learns the joint action-value $Q^{tot}(s_t, \boldsymbol{a}_t)$ via temporal-difference (TD) learning and performs credit assignment to obtain per-agent utilities $Q^i(\tau_t^i, a_t^i)$, which make decentralized policies, e.g., $\pi^i := \arg\max_{a_t^i \in \mathcal{A}^i} Q^i(\tau_t^i, \cdot)$.

### 3.2. Adversarial Attacks for MARL

To improve robustness against external perturbations, prior work in MARL has considered a range of adversarial attacks. A basic approach perturbs the state or observation by injecting noise, e.g., $\tilde{s}_t = s_t + \epsilon$ or $\tilde{\boldsymbol{o}}_t = \boldsymbol{o}_t + \epsilon$ (Lin et al., 2020), where the key design choice is how to construct $\epsilon$. Action attacks have also been widely studied. Under CTDE, a common formulation defines an attacker policy $\boldsymbol{\pi}_{\text{adv}}$ that selects perturbed actions to minimize learned utility estimates, e.g., $\tilde{a}_t^i \sim \pi_{\text{adv}}^i := \arg\min_{a^i \in \mathcal{A}^i} Q^i(\tau_t^i, a^i)$. For example, EGA (Yuan et al., 2023) diversifies value-minimizing attacks by learning critical timesteps for multiple random seeds, whereas the Wolfpack adversarial attack (Lee et al., 2025) sequentially targets agents to amplify disruption. In this paper, we consider both observation and action perturbations, but adopt an information-theoretic criterion that differs fundamentally from value-based objectives.

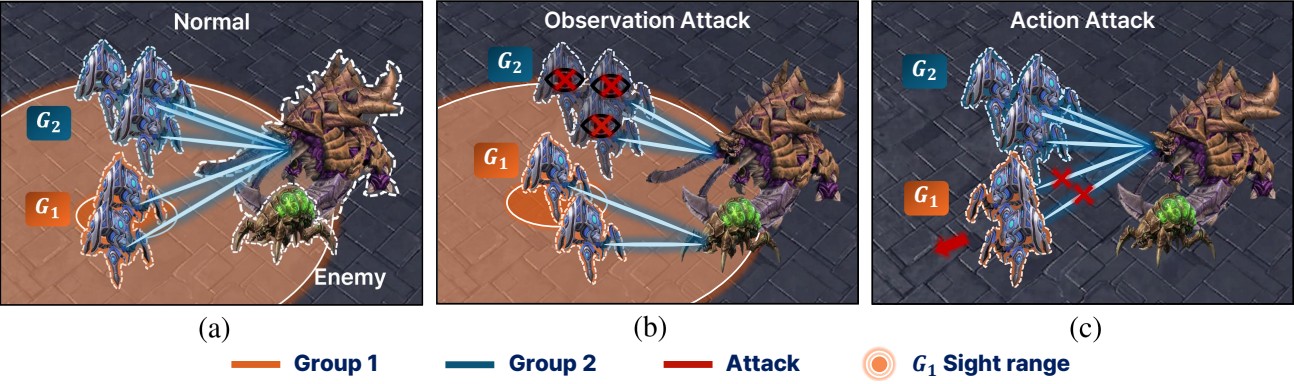

*Figure 1.* Illustration of the proposed interaction-breaking attack in StarCraft II. (a) Normal setting, where agents in $G_1$ and $G_2$ coordinate their attack on the same enemy. (b) Observation attack, where red crosses indicate masked observation components. (c) Action attack, where the red arrow denotes the perturbed action.

## 4. Methodology

### 4.1. Motivation for Interaction Breaking

In MARL, CTDE enables agents to learn coordinated strategies that solve multi-agent tasks efficiently, but this coupling can also make the learned policies brittle: when coordination partially fails, agents may face interaction patterns that were rarely encountered during training, resulting in severe performance degradation. Although several adversarial learning methods in MARL consider robustness under attacks (Lin et al., 2020; Yuan et al., 2023; Lee et al., 2025), they largely focus on simple perturbations or value-minimizing objectives. These formulations typically do not explicitly model how agents influence one another, and therefore may fail to capture attacks that intentionally break inter-agent relationships, under which coordination can collapse abruptly.

To study and mitigate this vulnerability, we propose an *interaction-breaking attack* that explicitly targets cross-agent influence. Concretely, we repeatedly partition agents into two groups, $G_1$ and $G_2$, quantify cross-group influence via MI, and construct observation and action attacks that minimize inter-group dependence. Fig. 1 provides an intuitive illustration of our attack. Fig. 1(a) shows the nominal setting, where $G_1$ and $G_2$ retain visibility and influence, enabling coordinated engagement. Fig. 1(b) illustrates our observation attack, which masks the components of $G_1$'s observations that are most informative about $G_2$, thereby reducing cross-group visibility. Fig. 1(c) illustrates our action attack, which perturbs actions to minimize MI and suppress cross-group influence, preventing $G_1$ from effectively coordinating with $G_2$. By applying our attacks across diverse group partitions, we expose policies to a spectrum of interaction failures that capture varied coordination-collapse scenarios. Finally, guided by our theoretical formulation, we aim to learn a robust MARL policy that remains effective even when coordination partially collapses.

### 4.2. Joint-Adversarial Dec-POMDP

In this section, we formalize our interaction-breaking attack as a joint adversary that perturbs both observations and actions. Specifically, an observation attacker $\boldsymbol{f}_{\text{adv}}$ produces perturbed observations $\tilde{\boldsymbol{o}}_t \sim \boldsymbol{f}_{\text{adv}}(\cdot \mid \boldsymbol{o}_t)$, and an action attacker $\boldsymbol{\pi}_{\text{adv}}$ generates perturbed actions $\tilde{\boldsymbol{a}}_t \sim \boldsymbol{\pi}_{\text{adv}}(s_t, \boldsymbol{a}_t)$, where $\boldsymbol{a}_t$ is sampled from the ego joint policy $\boldsymbol{\pi}$. With such an attacker that simultaneously targets observations and actions, the resulting process can be formulated as a Joint-Adversarial Decentralized POMDP (JA-Dec-POMDP).

**Definition 4.1** (Joint-Adversarial Dec-POMDP). Given a Dec-POMDP $\mathcal{M}$ and joint attackers $\boldsymbol{f}_{\text{adv}}$ and $\boldsymbol{\pi}_{\text{adv}}$, we define the *Joint-Adversarial Dec-POMDP* as $\mathcal{M}^J = \langle \mathcal{N}, \mathcal{S}, \mathcal{A}, P, \Omega, O, R, \gamma, \boldsymbol{f}_{\text{adv}}, \boldsymbol{\pi}_{\text{adv}} \rangle$. At each time step $t$, the observation attacker perturbs the original joint observation $\boldsymbol{o}_t$ by sampling $\tilde{\boldsymbol{o}}_t \sim \boldsymbol{f}_{\text{adv}}(\cdot \mid \boldsymbol{o}_t)$. Given $\tilde{\boldsymbol{o}}_t$, the ego joint policy selects an intermediate joint action $\hat{\boldsymbol{a}}_t \sim \boldsymbol{\pi}(\cdot \mid \tilde{\boldsymbol{o}}_t)$. The action attacker then outputs the executed joint action by sampling $\tilde{\boldsymbol{a}}_t \sim \boldsymbol{\pi}_{\text{adv}}(\cdot \mid s_t, \hat{\boldsymbol{a}}_t)$. The environment transitions as $s_{t+1} \sim P(\cdot \mid s_t, \tilde{\boldsymbol{a}}_t)$ and yields reward $r_t = R(s_t, \tilde{\boldsymbol{a}}_t, s_{t+1})$.

In the original Dec-POMDP $\mathcal{M}$, we define the value of a joint policy $\boldsymbol{\pi}$ as $V_{\boldsymbol{\pi}}(s_t) := \mathbb{E}_{\boldsymbol{\pi}}\left[\sum_{l=t}^{\infty} \gamma^{l-t} r_l \mid s_t\right]$. In the JA-Dec-POMDP, the joint attackers induce a perturbed decision process that can be equivalently viewed as the composition $\boldsymbol{\pi}_{\text{adv}} \circ \boldsymbol{\pi} \circ \boldsymbol{f}_{\text{adv}}$. Accordingly, we denote its value by $V_{\boldsymbol{\pi}_{\text{adv}} \circ \boldsymbol{\pi} \circ \boldsymbol{f}_{\text{adv}}}(s_t)$. As in standard policy optimization, one would like to find a policy that maximizes this value. However, directly optimizing over the composed policy is cumbersome. Instead, we reinterpret the joint attack as an environmental perturbation and perform policy optimization in the induced process, based on the following theorem.

**Theorem 4.2.** *Given a JA-Dec-POMDP $\mathcal{M}^J$ with joint attackers $\boldsymbol{f}_{\text{adv}}$ and $\boldsymbol{\pi}_{\text{adv}}$, there exists an induced Dec-POMDP $\tilde{\mathcal{M}} = \langle \mathcal{N}, \tilde{\mathcal{S}}, \mathcal{A}, \tilde{P}, \Omega, \tilde{O}, \tilde{R}, \gamma \rangle$, whose state*

*space, transition probability, observation, and reward functions explicitly account for the joint attack. Specifically,*

$$\tilde{\boldsymbol{o}}_t := \tilde{O}(s_t, i) = \boldsymbol{f}_{\text{adv}}(\cdot \mid O(s_t, i)), \ \ \hat{\boldsymbol{a}}_t \sim \boldsymbol{\pi}(\cdot|\tilde{\boldsymbol{o}}_t),$$
$$\tilde{P}(\tilde{s}_{t+1} \mid \tilde{s}_t, \hat{\boldsymbol{a}}_t) := P(s_{t+1} \mid s_t, \hat{\boldsymbol{a}}_t) \cdot \boldsymbol{\pi}_{\text{adv}}(\tilde{\boldsymbol{a}}_t \mid s_t, \hat{\boldsymbol{a}}_t),$$
$$\tilde{R}(\tilde{s}_t, \hat{\boldsymbol{a}}_t, \tilde{s}_{t+1}) := R(s_t, \tilde{\boldsymbol{a}}_t, s_{t+1}),$$

*where $\tilde{s}_t := (s_t, \tilde{\boldsymbol{a}}_t) \in \tilde{\mathcal{S}}$ with $\tilde{\boldsymbol{a}}_t \sim \boldsymbol{\pi}_{\text{adv}} \circ \boldsymbol{\pi} \circ \boldsymbol{f}_{\text{adv}}$. Then, for any joint policy $\boldsymbol{\pi}$ and all $s_t \in \mathcal{S}$, the state value $\tilde{V}_{\boldsymbol{\pi}}(s_t)$ of $\boldsymbol{\pi}$ in the Dec-POMDP $\tilde{\mathcal{M}}$ is equivalent to the state value $V_{\boldsymbol{\pi}_{\text{adv}} \circ \boldsymbol{\pi} \circ \boldsymbol{f}_{\text{adv}}}(s_t)$ in the JA-Dec-POMDP $\mathcal{M}^J$.*

*Proof.* Proof of Theorem 4.2 is provided in Appendix A.1.

By Theorem 4.2, the value in the JA-Dec-POMDP is equivalent to the value of the induced Dec-POMDP $\tilde{\mathcal{M}}$ with perturbed dynamics. Therefore, we can optimize the joint policy $\boldsymbol{\pi}$ in $\tilde{\mathcal{M}}$ using standard MARL algorithms under the perturbed transition dynamics, and the resulting policy preserves its performance under the JA-Dec-POMDP.

### 4.3. Interaction-Breaking Attack

As discussed in Section 4.1, we quantify cross-group influence using conditional MI and design an interaction-breaking attack based on it. We begin by partitioning the agent set $\mathcal{N}$ into two disjoint groups, $G_1 = \{i_1, \ldots, i_K\}$ and $G_2 = \mathcal{N} \setminus G_1$, where $K$ is the number of agents in $G_1$. For any per-agent variable $x^i$, we denote its restriction to $G_1$ by $\boldsymbol{x}^{G_1} := \{x^{i_1}, \ldots, x^{i_K}\}$. We then define the influence of $G_2$'s joint action on $G_1$ given the shared history $\boldsymbol{\tau}_t$ via conditional MI as follows.

$$\mathcal{I}\big(\boldsymbol{o}_{t+1}^{G_1}, \boldsymbol{a}_t^{G_1}; \boldsymbol{a}_t^{G_2} \mid \boldsymbol{\tau}_t\big) = \underbrace{\mathcal{I}\big(\boldsymbol{o}_{t+1}^{G_1}; \boldsymbol{a}_t^{G_2} \mid \boldsymbol{a}_t^{G_1}, \boldsymbol{\tau}_t\big)}_{\textit{observation-level MI}}$$
$$+ \underbrace{\mathcal{I}\big(\boldsymbol{a}_t^{G_1}; \boldsymbol{a}_t^{G_2} \mid \boldsymbol{\tau}_t\big)}_{\textit{action-level MI}}, \quad (1)$$

where $\mathcal{I}(X; Y|Z) := H(X|Z) - H(X|Y, Z)$ denotes conditional MI, which quantifies how much information $Y$ provides about $X$ given $Z$, and $H(X) := \mathbb{E}[-\log p(X)]$ denotes entropy. Here, the action- and observation-level MI terms follow from the chain rule of MI. The *observation-level MI* measures how sensitive the observations of agents in $G_1$ are to the actions of $G_2$. For example, when agents in $G_2$ enter or leave the field of view of $G_1$, leading to large changes in $\boldsymbol{o}_{t+1}^{G_1}$. In contrast, the *action-level MI* measures how tightly the actions of $G_1$ are coupled with those of $G_2$; it becomes large when the two groups coordinate, making $\boldsymbol{a}_t^{G_1}$ highly informative about $\boldsymbol{a}_t^{G_2}$. Since these two terms quantify cross-group influence, we design a joint attacker that breaks interaction by minimizing them separately: $\boldsymbol{f}_{\text{adv}}$ targets the observation-level MI, while $\boldsymbol{\pi}_{\text{adv}}$ targets the action-level MI, as defined below.

**Observation attacker.** The observation attacker aims to reduce the observation-level MI term by masking a subset of dimensions in the observations of agents in $G_1$. We define a masking attacker $\tilde{\boldsymbol{o}}_t = \boldsymbol{f}_{\text{adv}}(\boldsymbol{o}_t) := \{\boldsymbol{m}^{G_1}(\boldsymbol{o}_t^{G_1}; \boldsymbol{D}^{G_1}), \boldsymbol{o}_t^{G_2}\}$, where the agent ordering is assumed to be aligned. Here, $m^i(o_t^i; D^i)$ applies a zero-forcing mask to the dimensions indexed by $D^i$: for the $d$-th component $o_{d,t}^i$ of $o_t^i$, it outputs $\tilde{o}_{d,t}^i = 0$ if $d \in D^i$ and $\tilde{o}_{d,t}^i = o_{d,t}^i$ otherwise. We control the masking strength by fixing the per-agent budget $|D^i| = L$ for each $i \in G_1$. This design is motivated by the data-processing inequality, $\mathcal{I}(f(X); Y|Z) \leq \mathcal{I}(X; Y|Z)$, which implies that applying any mapping $f$ cannot increase conditional MI; in particular, a deterministic mask typically decreases it by removing variation along the masked dimensions.

Ideally, we would select $\boldsymbol{D}^{G_1}$ to maximally reduce the resulting observation-level MI. However, evaluating MI for all possible masks is computationally prohibitive, and the cost further increases when the partition changes. We therefore adopt an efficient alternative based on the following lemma, which upper bounds the group-wise MI by a sum of dimension-wise terms.

**Lemma 4.3.** *Given a masking attacker $\boldsymbol{m}^{G_1}$, the group-wise observation-level MI can be upper-bounded as*

$$\mathcal{I}(\tilde{\boldsymbol{o}}_{t+1}^{G_1}; \boldsymbol{a}_t^{G_2} \mid \boldsymbol{a}_t^{G_1}, \boldsymbol{\tau}_t)$$
$$\leq \sum_{i \in G_1} \sum_{d \notin D^i} \underbrace{\sum_{j \in G_2} \mathcal{I}(o_{d,t+1}^i; a_t^j \mid a_t^i, \boldsymbol{\tau}_t)}_{\textit{dimension-wise MI}} + \mathcal{R}(G_1; G_2), \quad (2)$$

*where $d_o$ denotes the observation dimension and $\mathcal{R}(G_1; G_2)$ denotes a group redundancy term.*

*Proof.* A proof of Lemma 4.3 is provided in Appendix A.2.

By Lemma 4.3, the group-wise MI decomposes into the dimension-wise MI and a group-redundancy term. This redundancy term is known to decay much faster than the MI term (Kubkowski et al., 2021), similar to higher-order residuals in a Taylor expansion. We also confirm this empirically in Appendix F.1, where the redundancy term is consistently negligible relative to the MI term. Accordingly, we approximate the group-wise MI using the dimension-wise MI in practice. Under this approximation, for each agent $i$, minimizing $\sum_{d \notin D^i} \sum_{j \in G_2} \mathcal{I}(o_{d,t+1}^i; a_t^j \mid a_t^i, \boldsymbol{\tau}_t)$ is equivalent to masking the $L$ dimensions with the largest dimension-wise MI. We therefore define the interaction-breaking observation attacker as follows.

**Definition 4.4** (Interaction-Breaking Observation Attacker). An *interaction-breaking observation attacker* $\boldsymbol{f}_{\text{adv}}^{\text{IB}}$ selects, for each $i \in G_1$, a set $D^{i,*}$ of $L$ observation dimensions that maximize the dimension-wise observation-level MI, i.e.,

$$D^{i,*} = \underset{D^i : |D^i| = L}{\arg\max} \sum_{d \in D^i} \sum_{j \in G_2} \mathcal{I}\big(o_{d,t+1}^i; a_t^j \mid a_t^i, \boldsymbol{\tau}_t\big).$$

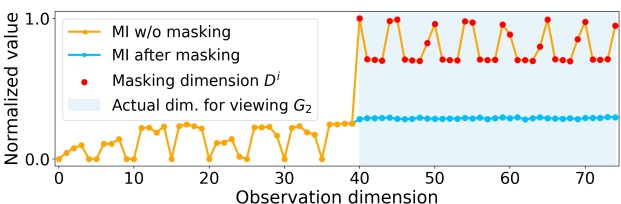

*Figure 2.* Dimension-wise MI for the $G_1$ agent in the StarCraft II scenario. MI values are normalized to $[0, 1]$. Here, $|G_1| = 1$ and $|G_2| = 7$, and we set $L = 5 \times |G_2|$ to match the number of $G_1$ observation dimensions used to observe $G_2$ agents.

It then masks $\boldsymbol{D}^{G_1,*} := \{D^{i,*}\}_{i \in G_1}$ via

$$\tilde{\boldsymbol{o}}_{t+1} = \boldsymbol{f}^{IB}_{adv}(\boldsymbol{o}_{t+1}) := \left\{ \boldsymbol{m}^{G_1}(\boldsymbol{o}^{G_1}_{t+1}; \boldsymbol{D}^{G_1,*}),\ \boldsymbol{o}^{G_2}_{t+1} \right\}.$$

This definition only requires computing dimension-wise MI scores once per agent and observation dimension. When the grouping changes, the attacker is instantiated by aggregating the precomputed scores, avoiding repeated MI estimation and substantially reducing computational cost.

To illustrate the effectiveness of the proposed attacker, Fig. 2 reports the dimension-wise MI of a representative agent $i \in G_1$ before and after masking, for a fixed $G_2$, together with the selected index set $D^{i,*}$ and the ground-truth indices corresponding to dimensions that encode the presence of $G_2$ in $G_1$'s observation. The result shows that high-MI dimensions coincide with those that actually observe $G_2$, indicating that our criterion identifies influential observation components. Moreover, masking sets the MI of the selected dimensions to zero, substantially reducing the overall MI. Operationally, this prevents agents in $G_1$ from observing $G_2$, enabling efficient interaction breaking.

**Action attacker.** We design an action attacker that directly minimizes the action-level MI. Given the perturbed observations $\tilde{\boldsymbol{o}}_t$, the ego joint policy first samples an intermediate joint action $\hat{\boldsymbol{a}}_t \sim \boldsymbol{\pi}(\cdot \mid \tilde{\boldsymbol{o}}_t)$. The attacker then perturbs this action as defined below.

**Definition 4.5** (Interaction-Breaking Action Attacker). Given an intermediate joint action $\hat{\boldsymbol{a}}_t$, let $\tilde{\boldsymbol{a}}^{\min,G_1}_t$ denote the joint action of agents in $G_1$ that minimizes the action-level MI, defined as $\tilde{\boldsymbol{a}}^{\min,G_1}_t := \arg\min_{\boldsymbol{a}^{G_1}_t} \mathcal{I}(\boldsymbol{a}^{G_1}_t; \hat{\boldsymbol{a}}^{G_2}_t \mid \boldsymbol{\tau}_t)$. An *interaction-breaking action attacker* $\boldsymbol{\pi}^{IB}_{adv}$ selects the joint action $\tilde{\boldsymbol{a}}_t$ as

$$\tilde{\boldsymbol{a}}_t = \begin{cases} \langle \tilde{\boldsymbol{a}}^{\min,G_1}_t,\ \hat{\boldsymbol{a}}^{G_2}_t \rangle, & \text{with probability } P_{act}, \\ \hat{\boldsymbol{a}}_t, & \text{with probability } 1 - P_{act}, \end{cases}$$

where we assume the agent ordering is aligned, and $P_{act}$ denotes the probability of applying the action attack.

As a result, the proposed action attack suppresses $G_1$ actions that would otherwise influence $G_2$, thereby disrupting cross-group coordination and influence.

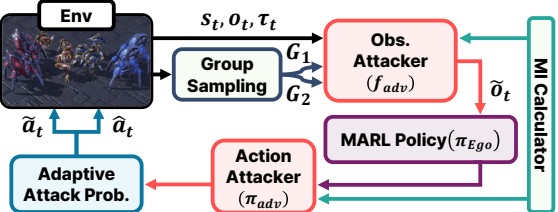

*Figure 3.* Illustration of the proposed IBAL framework.

Finally, implementing the proposed joint attackers $\boldsymbol{f}^{IB}_{adv}$ and $\boldsymbol{\pi}^{IB}_{adv}$ requires estimating the associated MI terms. Following prior work (Jaques et al., 2019), we adopt a model-based approach. For the observation-level MI, we adopt CLUB (Cheng et al., 2020) to estimate dimension-wise MI, enabling efficient scoring over observation dimensions. For the action-level MI, we formulate the objective via the Kullback–Leibler divergence and directly estimate group-wise MI with a shared model, without dimension-wise decomposition. Details of MI estimation are provided in Appendix B.

### 4.4. IBAL Framework and Implementation

With the interaction-breaking attackers, we can instantiate the JA-Dec-POMDP $\mathcal{M}^J$ and its induced Dec-POMDP $\tilde{\mathcal{M}}$. Based on Theorem 4.2, we propose the *interaction-breaking adversarial learning* (IBAL) framework, which learns a MARL policy under CTDE on $\tilde{\mathcal{M}}$ with perturbed dynamics and yields a policy that remains robust in $\mathcal{M}^J$.

In implementation, to expose the learner to attacks from diverse partitions, we randomly sample the groups at the beginning of each episode. Specifically, we sample $k \sim \text{Unif}(\{0, \ldots, K\})$ with $K \leq n/2$, draw $G_1 \sim \text{Unif}(\{G \subset \mathcal{N} : |G| = k\})$, and set $G_2 = \mathcal{N} \setminus G_1$. This keeps $G_1$ smaller than $G_2$ to avoid overly strong attacks. For the observation attack, although the formulation targets occluding $G_1$'s view of $G_2$, we apply the masking symmetrically so that both groups are mutually occluded. For the action attack, instead of a fixed attack probability, we use an adaptive schedule that gradually increases attack strength. Concretely, we sample $P_{act} \sim \text{Unif}\left(\frac{1}{K}, P^{max}_{act}\right)$ and update $P^{max}_{act}$ by

$$P^{max}_{act} \leftarrow \begin{cases} \min(1, \alpha P^{max}_{act}), & \text{if } \bar{\sigma} \geq \eta, \\ P^{max}_{act}, & \text{otherwise,} \end{cases}$$

where $\bar{\sigma}$ denotes the average success rate, $\alpha > 1$ controls the growth rate, and $\eta$ is a success rate threshold. If success rates are unavailable, we use the average episodic return. Here, we set the minimum attack probability to $1/K$ to ensure a nontrivial level of attack. While IBAL is broadly applicable to CTDE algorithms, we primarily instantiate it with QMIX (Rashid et al., 2020b). The overall IBAL framework is illustrated in Fig. 3 and summarized in Algorithm 1, with additional implementation details provided in Appendix B.

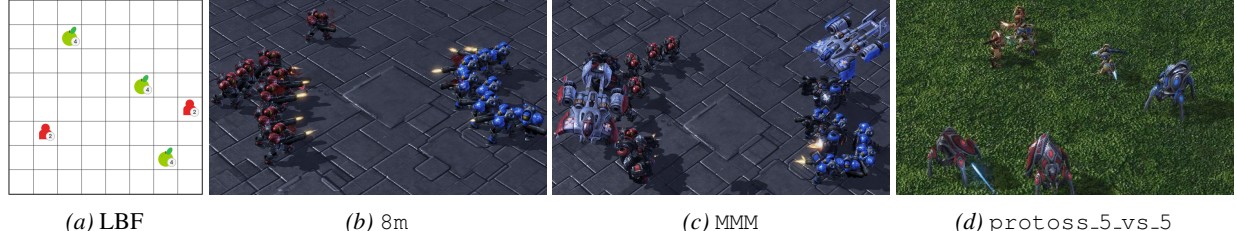

*(a)* LBF        *(b)* 8m        *(c)* MMM        *(d)* protoss_5_vs_5

*Figure 4.* Visualization of MARL benchmarks: (a) Level-Based Foraging (LBF), (b) and (c) SMAC, and (d) SMACv2.

---

**Algorithm 1** IBAL framework

> **Initialize:** $Q^{\text{tot}}$, MI estimators, $P_{\text{act}}$
> **for** each training iteration **do**
>     Randomly partition agents into $G = \{G_1, G_2\}$
>     **for** each environment step $t$ **do**
>        Apply observation attack: $\tilde{\boldsymbol{o}}_t \sim \boldsymbol{f}_{\text{adv}}^G(\cdot \mid \boldsymbol{o}_t)$,
>        Sample ego joint action: $\hat{\boldsymbol{a}}_t \sim \boldsymbol{\pi}(\cdot \mid \tilde{\boldsymbol{o}}_t)$
>        Apply action attack with probability $P_{\text{act}}$:
>        $\tilde{\boldsymbol{a}}_t \sim \boldsymbol{\pi}_{\text{adv}}(\cdot \mid s_t, \hat{\boldsymbol{a}}_t)$.
>        Execute $\tilde{\boldsymbol{a}}_t$ and store the transition
>     **end for**
>     Update $Q^{\text{tot}}$ using a CTDE algorithm
>     Update the MI estimators
>     Update the action attack probability $P_{\text{act}}$
> **end for**

---

## 5. Experiments

### 5.1. Experimental Setup

In this section, we compare the robustness of the proposed IBAL and prior robust MARL methods on the StarCraft II Multi-Agent Challenge (SMAC) (Samvelyan et al., 2019), which requires cooperative decision-making among ally units to defeat enemy units. As illustrated in Fig. 4, we consider six scenarios with diverse unit compositions and formations: 3m, 3s_vs_3z, 2s3z, 8m, 1c3s5z, and MMM. We also conduct additional experiments on Level-Based Foraging (LBF) (Papoudakis et al., 2020), a simpler cooperative environment, and SMACv2 (Ellis et al., 2023), a more stochastic extension of SMAC. Further details, including task descriptions and reward configurations, are provided in Appendix C. All performance graphs report the mean over 5 random seeds (solid line) with standard deviation (shaded region). Our code is available at https://sunwoolee0504.github.io/IBAL.

For evaluation, we consider a suite of adversarial attacks from prior works and non-parametric perturbations that naturally arise in SMAC, and compare MARL baselines accordingly, as summarized below:

**Attacker Baselines.** We consider adversarial attacks that perturb observations and actions: **Natural (Nat.)**, the no-attack setting; **Rand.**, which applies random perturbations to observations and actions; and **FGSM** (Goodfellow et al., 2014), which considers a gradient-based observation attack. We also include two value-minimizing action attacks, **EGA** (Yuan et al., 2023) and **Wolfpack Attack (Wolf.)** (Lee et al., 2025), introduced in Section 3.2, as well as **Ours**, the proposed interaction-breaking attack.

**Non-parametric Perturbations.** We further evaluate robustness under natural non-parametric perturbations. We consider **Dis-$\ell$**, which disables $\ell$ randomly selected ally units, and **HP-$h$**, which reduces ally units' initial health by $h\%$. **Dis-$\ell$** tests coordination under ally-unit absence, while **HP-$h$** evaluates robustness under reduced initial health.

**Robust MARL baselines.** We compare QMIX-based robust MARL methods trained under different attacks: **Vanilla QMIX** (Rashid et al., 2020b) without adversarial training; **Rand-Obs** and **Rand-Act**, trained under random observation and action attacks; **FGSM** (Goodfellow et al., 2014), trained under FGSM-based observation attacks; **ATLA** (Zhang et al., 2021a), trained against an RL-learned observation attacker; **ERNIE** (Bukharin et al., 2023) with regularization-based robust training; **ROMANCE** (Yuan et al., 2023), trained against EGA; **WALL** (Lee et al., 2025), trained against Wolfpack attack; and **IBAL (Ours)**, trained against the proposed interaction-breaking attack. All policies are trained for 10M timesteps, initialized from a QMIX model pretrained for 1M timesteps. Robustness is evaluated under unseen attacks generated with different random seeds than those used in training. For IBAL, the maximum group size $K$ is selected via hyperparameter search during training. Detailed descriptions of all baselines and additional experimental settings are provided in Appendix D.

### 5.2. Performance Comparison

We report the mean test win rate over 5 seeds under adversarial attacks and non-parametric perturbations in Fig. 5 and Fig. 6, respectively. Full result tables with standard deviations and learning curves for the considered algorithms are provided in Appendix E. Under adversarial attacks, IBAL consistently outperforms prior robust MARL methods, indicating improved robustness across a broader range of attacks. Notably, methods such as FGSM and WALL often perform

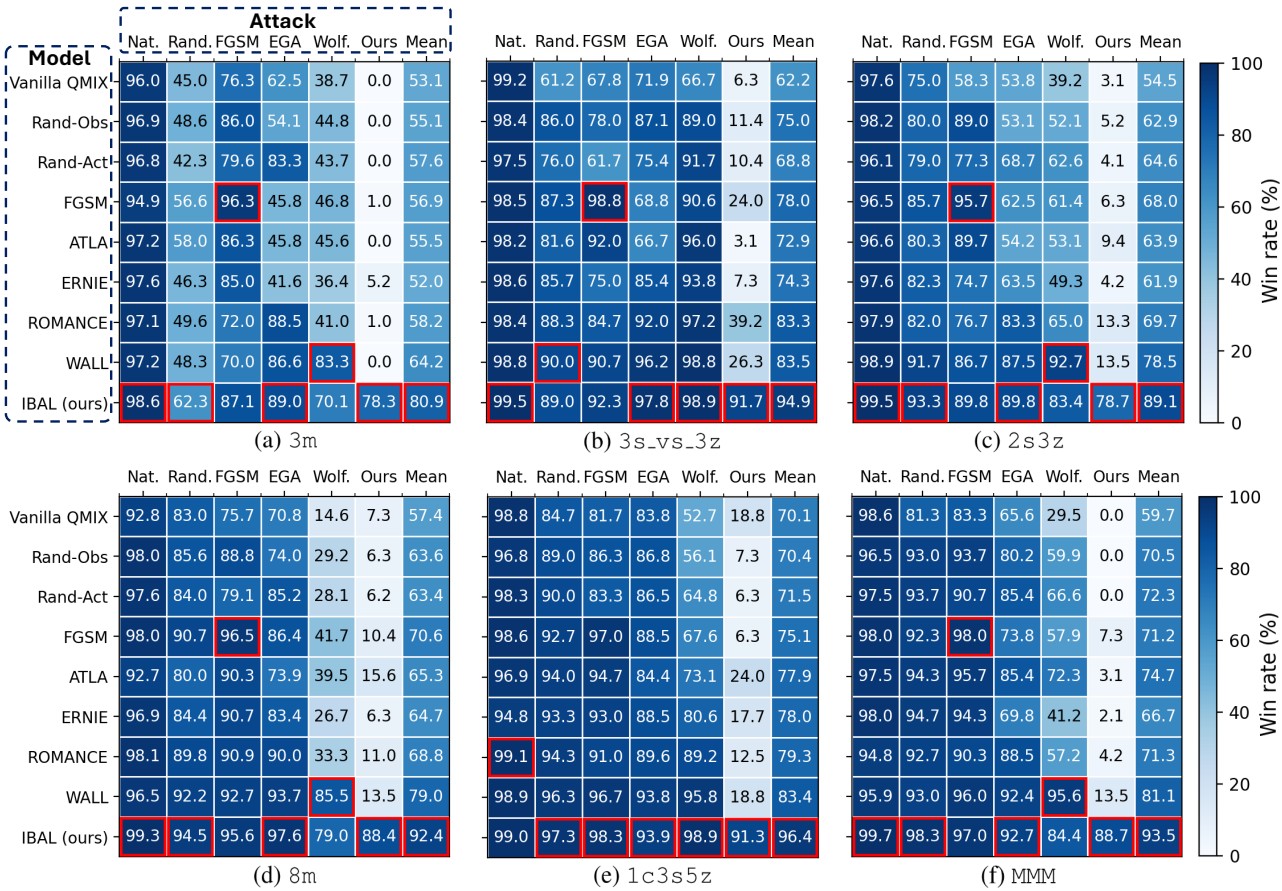

*Figure 5.* Average test win rate under various adversarial attacks.

well against their own attacks, yet their performance drops sharply under our interaction-breaking attack. In contrast, IBAL not only defends effectively against the proposed attack, but also generalizes well to other observation and action attacks. In particular, on 1c3s5z, IBAL attains high win rates across all tested attacks. These results suggest that mutual-information-driven interaction breaking is a critical failure mode for cooperation, and training against it yields policies with stronger robustness and generalization.

For non-parametric perturbations, we consider Dis-$\ell$ with $\ell \in \{1, 2\}$, and HP-$h$ with $h \in \{10, 15, 20\}$. We report results for $h = 15$ in the main text, and provide remaining results in Appendix E.2. Under these perturbations, IBAL shows a clear advantage over all baselines. In the Dis-$\ell$ setting, most methods degrade sharply when ally units are disabled, revealing brittle coordination, whereas IBAL remains effective, as training over diverse group partitions exposes the policy to partial interaction failures and encourages adaptive coordination. A similar trend holds for HP-$h$: while other methods retain some performance, IBAL achieves substantially higher win rates. Overall, these results suggest that IBAL is robust to non-parametric perturbations

by learning to operate under disrupted group interactions.

We further assess the generality of IBAL on additional benchmarks and another CTDE backbone. On LBF and SMACv2, the proposed interaction-breaking attack is more damaging than conventional attacks because it directly disrupts cooperation, while IBAL remains robust not only under the proposed attack but also across existing attack settings. In particular, on SMACv2, despite the increased stochasticity that lowers natural performance compared to the original SMAC scenarios, IBAL achieves stronger natural performance than the baselines, suggesting robustness to stochasticity as well as explicit attacks. Moreover, because IBAL does not rely on value information or the IGM property, it can be extended beyond value-based methods; results with MAPPO show that IBAL also improves robustness for policy-gradient-based CTDE methods. Detailed results are provided in Appendix E.3 and Appendix E.4.

### 5.3. Further Analysis of IBAL

To analyze the behavior of the proposed attacker and how IBAL responds, Fig. 7 presents trajectory analyses in 8m

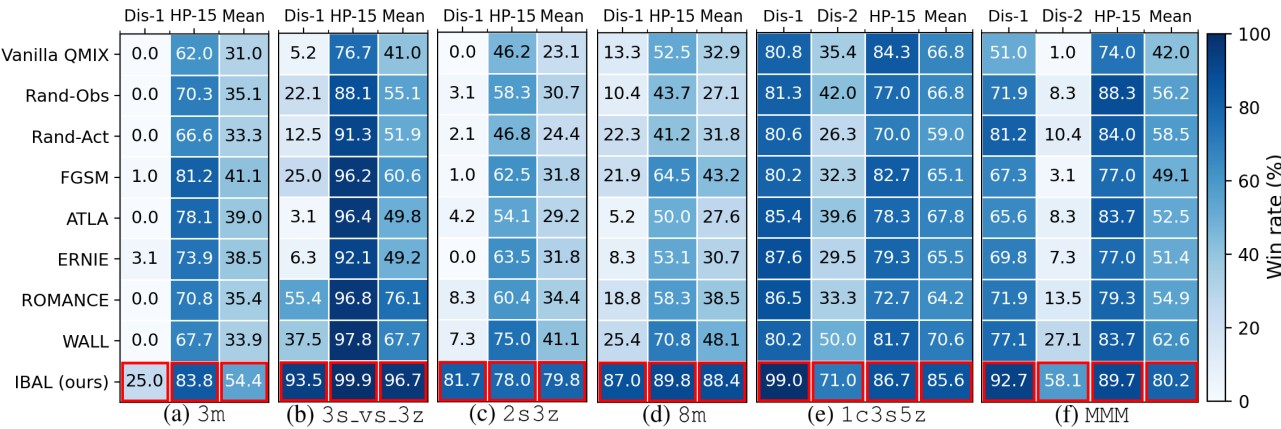

| | Dis-1 | HP-15 | Mean | Dis-1 | HP-15 | Mean | Dis-1 | HP-15 | Mean | Dis-1 | HP-15 | Mean | Dis-1 | Dis-2 | HP-15 | Mean | Dis-1 | Dis-2 | HP-15 | Mean |
|---|---|---|---|---|---|---|---|---|---|---|---|---|---|---|---|---|---|---|---|---|
| Vanilla QMIX | 0.0 | 62.0 | 31.0 | 5.2 | 76.7 | 41.0 | 0.0 | 46.2 | 23.1 | 13.3 | 52.5 | 32.9 | 80.8 | 35.4 | 84.3 | 66.8 | 51.0 | 1.0 | 74.0 | 42.0 |
| Rand-Obs | 0.0 | 70.3 | 35.1 | 22.1 | 88.1 | 55.1 | 3.1 | 58.3 | 30.7 | 10.4 | 43.7 | 27.1 | 81.3 | 42.0 | 77.0 | 66.8 | 71.9 | 8.3 | 88.3 | 56.2 |
| Rand-Act | 0.0 | 66.6 | 33.3 | 12.5 | 91.3 | 51.9 | 2.1 | 46.8 | 24.4 | 22.3 | 41.2 | 31.8 | 80.6 | 26.3 | 70.0 | 59.0 | 81.2 | 10.4 | 84.0 | 58.5 |
| FGSM | 1.0 | 81.2 | 41.1 | 25.0 | 96.2 | 60.6 | 1.0 | 62.5 | 31.8 | 21.9 | 64.5 | 43.2 | 80.2 | 32.3 | 82.7 | 65.1 | 67.3 | 3.1 | 77.0 | 49.1 |
| ATLA | 0.0 | 78.1 | 39.0 | 3.1 | 96.4 | 49.8 | 4.2 | 54.1 | 29.2 | 5.2 | 50.0 | 27.6 | 85.4 | 39.6 | 78.3 | 67.8 | 65.6 | 8.3 | 83.7 | 52.5 |
| ERNIE | 3.1 | 73.9 | 38.5 | 6.3 | 92.1 | 49.2 | 0.0 | 63.5 | 31.8 | 8.3 | 53.1 | 30.7 | 87.6 | 29.5 | 79.3 | 65.5 | 69.8 | 7.3 | 77.0 | 51.4 |
| ROMANCE | 0.0 | 70.8 | 35.4 | 55.4 | 96.8 | 76.1 | 8.3 | 60.4 | 34.4 | 18.8 | 58.3 | 38.5 | 86.5 | 33.3 | 72.7 | 64.2 | 71.9 | 13.5 | 79.3 | 54.9 |
| WALL | 0.0 | 67.7 | 33.9 | 37.5 | 97.8 | 67.7 | 7.3 | 75.0 | 41.1 | 25.4 | 70.8 | 48.1 | 80.2 | 50.0 | 81.7 | 70.6 | 77.1 | 27.1 | 83.7 | 62.6 |
| IBAL (ours) | 25.0 | 83.8 | 54.4 | 93.5 | 99.9 | 96.7 | 81.7 | 78.0 | 79.8 | 87.0 | 89.8 | 88.4 | 99.0 | 71.0 | 86.7 | 85.6 | 92.7 | 58.1 | 89.7 | 80.2 |
| | (a) 3m | | | (b) 3s_vs_3z | | | (c) 2s3z | | | (d) 8m | | | (e) 1c3s5z | | | | (f) MMM | | | |

*Figure 6.* Average test win rate under various non-parametric perturbations.

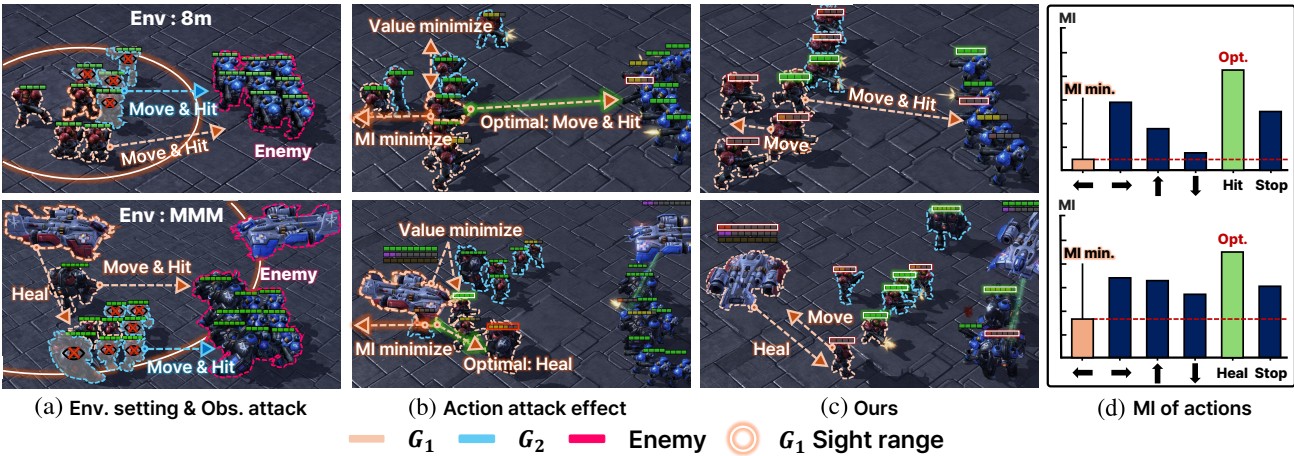

(a) **Env. setting & Obs. attack**   (b) **Action attack effect**   (c) **Ours**   (d) **MI of actions**

$G_1$   $G_2$   Enemy   $G_1$ Sight range

*Figure 7.* Trajectory analysis of the interaction-breaking attack and IBAL policies in 8m and MMM tasks.

and MMM: **(a)** the group partition and observation attack, **(b)** a comparison of the optimal action with value-minimizing and MI-minimizing actions, **(c)** behaviors learned by IBAL, and **(d)** MI scores over $G_1$ agents' actions. In **(a)**, the attacker masks observation components corresponding to $G_2$, effectively removing cross-group visibility, as in Fig. 2. In **(b)**, the optimal behavior in 8m is for marines to move forward and attack, while in MMM the medivac heals low-health allies. Guided by the action-wise MI in **(d)**, our attacker selects MI-minimizing actions that retreat to reduce influence on $G_2$ in both scenarios. In contrast, value-minimizing attacks often induce oscillatory movements due to QMIX's monotonic mixing, which can yield unreliable value estimates for non-optimal actions (Rashid et al., 2020a). In comparison, the proposed MI-based attacker is agnostic to the QMIX structure and directly minimizes an influence objective, yielding a more consistent strategy that effectively disrupts cooperation. Consequently, IBAL remains robust to diverse value-minimizing attacks, whereas baselines trained only against them struggle under interaction breaking.

Finally, **(c)** highlights how IBAL adapts under repeated exposure to diverse partitions and interaction-breaking attacks. In 8m, when an ally becomes vulnerable under attack, a healthier ally advances to hold the front line. In MMM, when the medivac is driven away and stops healing, low-health allies move toward it to receive heals and then return to the fight. These behaviors rarely emerge under standard training but arise naturally when agents must operate under frequent coordination failures. Overall, IBAL learns effective group-wise coordination under interaction disruption, resulting in stronger robustness under diverse attacks and perturbations. Additional analysis on how IBAL responds to Dis-$\ell$ is provided in Appendix F.2.

### 5.4. Ablation Study

To assess the contribution of each component, we conduct several ablations. In the main text, we report component-wise evaluations and analyze sensitivity to the maximum group size $K$. In Appendix F.3, we provide additional anal-

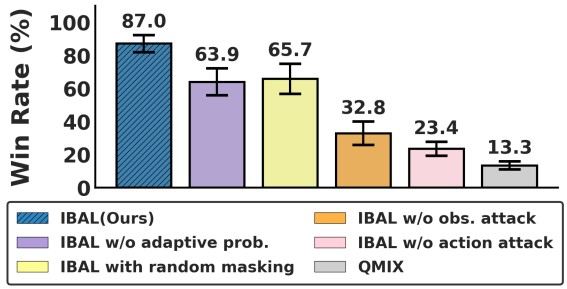

*Figure 8.* Component evaluation.

*Table 1.* Ablation on observation attack variants.

| Scenario | Variant | Win rate (%) |
|---|---|---|
| | IBAL | **88.4 ± 3.30** |
| 8m | IBAL with Gaussian noise | 78.1 ± 13.3 |
| | IBAL with FGSM | 38.5 ± 7.40 |
| | IBAL | **88.7 ± 7.23** |
| MMM | IBAL with Gaussian noise | 71.9 ± 22.1 |
| | IBAL with FGSM | 77.6 ± 3.60 |

yses on the masking budget $L$, the minimum attack probability, and computational complexity.

**Component Evaluation.** To quantify each component's contribution, we evaluate ablations on 8m under Dis-1. We compare **IBAL (Ours)** with four variants: **IBAL w/o adaptive prob.**, which fixes the action-attack probability to $P_{\text{act}} = 1/K$; **IBAL with random masking**, which masks $L$ dimensions uniformly at random instead of using the MI criterion; **IBAL w/o obs. attack**, which uses only the MI-minimizing action attack; and **IBAL w/o action attack**, which uses only observation masking. Fig. 8 reports test win rates. The result shows that removing any component degrades performance, indicating that each contributes to robustness. In particular, omitting either the observation or action attack causes a substantial drop, suggesting that both are necessary to capture interaction breaking. In addition, random masking and a fixed attack probability still yield gains, but MI-guided masking and the adaptive schedule are essential for better robustness.

**Observation Attack Variants.** We further analyze different observation attack variants applied to the selected dimensions $\boldsymbol{D}^{G_1}$. In the default IBAL, we apply zero-masking because it directly removes cross-group information. We compare this with Gaussian noise and FGSM-based observation attacks on the same selected dimensions, and evaluate these variants on 8m and MMM under Dis-1. As shown in Table 1, the default IBAL with zero-masking achieves the strongest robustness. This suggests that completely removing selected cross-group information is more effective for inducing interaction breaking than adding noisy or adversar-

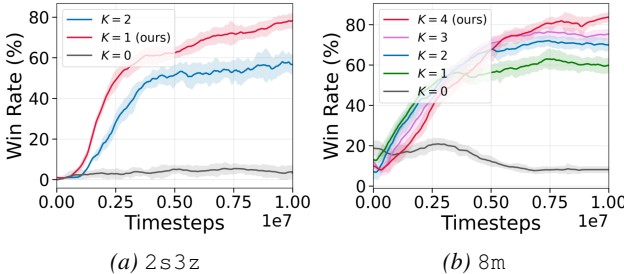

*(a)* 2s3z  *(b)* 8m

*Figure 9.* Analysis on the maximum group size $K$.

ial signals, which may still preserve residual mutual information and thus be less effective at disrupting inter-agent coordination.

**Maximum Group Size $K$.** We analyze the effect of the maximum attacked group size $K$ for $G_1$. To avoid overly strong attacks, we restrict $K \leq n/2$ so that $G_2$ remains at least as large as $G_1$. Fig. 9 reports results on 2s3z and 8m under Dis-1, where the effect of $K$ is most pronounced. When $K = 0$, no attack is applied, and performance matches Vanilla QMIX. As $K$ increases, robustness generally improves, but overly large $K$ can make the attack too severe and hinder learning a robust policy. We find that $K = 1$ works best on 2s3z, while 8m remains stable up to $K = 4$, which we use as the default setting.

## 6. Limitation

While IBAL achieves strong empirical performance, it has limitations. First, it introduces additional hyperparameters, such as the maximum group size $K$ and the masking budget $L$. In practice, we provide guideline ranges based on hyperparameter search, and we find performance is not overly sensitive within these ranges. Second, IBAL incurs extra computational overhead due to MI estimation. As analyzed in Appendix F.4, this overhead is moderate, and we find it reasonable given the corresponding performance gains.

## 7. Conclusion

In this paper, we proposed an interaction-breaking attack that minimizes cross-group influence under an MI criterion and introduced IBAL for learning policies robust to such disruptions. The attacker combines an observation attack that removes influential cross-group signals with an action attack that stochastically selects MI-minimizing actions, and we provide a theoretical justification for learning robust policies under the induced perturbed dynamics. Experiments show that IBAL improves robustness and generalization over prior methods across diverse attacks and perturbations.

## Acknowledgment

This work was supported partly by the Institute of Information & Communications Technology Planning & Evaluation (IITP) grant funded by the Korea government (MSIT) (No. RS-2022-II220469, Development of Core Technologies for Task-oriented Reinforcement Learning for Commercialization of Autonomous Drones), (IITP-2025-RS-2022-00156361, Innovative Human Resource Development for Local Intellectualization program), and (No. RS-2020-II201336, Artificial Intelligence Graduate School Support (UNIST)), and partly by the National Research Foundation of Korea (NRF) grant funded by the Korea government (MSIT) (No. RS-2025-23523191, LLM-Based Multi-Agent Reinforcement Learning for End-to-End Large Autonomous Swarm Control).

## Impact Statement

This paper presents work whose goal is to advance the field of Machine Learning. There are many potential societal consequences of our work, none which we feel must be specifically highlighted here.

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

# A. Missing Proofs

### A.1. Proof of Theorem 4.2

To prove Theorem 4.2, we first express the state-value function $V_{\boldsymbol{\pi}_{\mathrm{adv}}\circ\boldsymbol{\pi}\circ\boldsymbol{f}_{\mathrm{adv}}}$ of the JA-Dec-POMDP in Bellman equation form in Lemma A.1.

**Lemma A.1.** *Given a JA-Dec-POMDP $\mathcal{M}^J = \langle \mathcal{N}, \mathcal{S}, \mathcal{A}, P, \Omega, O, R, \gamma, \boldsymbol{f}_{\mathrm{adv}}, \boldsymbol{\pi}_{\mathrm{adv}} \rangle$ and an ego policy $\boldsymbol{\pi}$, the state-value function $V_{\boldsymbol{\pi}_{\mathrm{adv}}\circ\boldsymbol{\pi}\circ\boldsymbol{f}_{\mathrm{adv}}}$ satisfies*

$$V_{\boldsymbol{\pi}_{\mathrm{adv}}\circ\boldsymbol{\pi}\circ\boldsymbol{f}_{\mathrm{adv}}}(s_t) = \mathbb{E}_{\tilde{\boldsymbol{a}}_t \sim \boldsymbol{\pi}_{\mathrm{adv}}\circ\boldsymbol{\pi}\circ\boldsymbol{f}_{\mathrm{adv}}}\Big[\mathbb{E}_{s_{t+1}\sim P(\cdot|s_t,\tilde{\boldsymbol{a}}_t)}\big[R(s_t,\tilde{\boldsymbol{a}}_t,s_{t+1}) + \gamma V_{\boldsymbol{\pi}_{\mathrm{adv}}\circ\boldsymbol{\pi}\circ\boldsymbol{f}_{\mathrm{adv}}}(s_{t+1})\big]\Big]. \tag{3}$$

*Proof.* Starting from the definition of the state-value function $V(s_t) = \mathbb{E}_{\boldsymbol{\pi}}\big[\sum_{l=t}^{\infty} \gamma^{l-t} r_l \,\big|\, s_t\big]$, we can express the value induced by the composed policy $\boldsymbol{\pi}_{\mathrm{adv}} \circ \boldsymbol{\pi} \circ \boldsymbol{f}_{\mathrm{adv}}$ as follows:

$$\begin{aligned}
V_{\boldsymbol{\pi}_{\mathrm{adv}}\circ\boldsymbol{\pi}\circ\boldsymbol{f}_{\mathrm{adv}}}(s_t) &= \mathbb{E}_{\boldsymbol{\pi}_{\mathrm{adv}}\circ\boldsymbol{\pi}\circ\boldsymbol{f}_{\mathrm{adv}}}\left[\sum_{l=t}^{\infty} \gamma^{l-t} r_l \,\big|\, s_t\right] \\
&= \mathbb{E}_{\boldsymbol{\pi}_{\mathrm{adv}}\circ\boldsymbol{\pi}\circ\boldsymbol{f}_{\mathrm{adv}}}\left[r_{t+1} + \gamma r_{t+2} + \gamma^2 r_{t+3} + \cdots \,\big|\, s_t\right] \\
&= \mathbb{E}_{\boldsymbol{\pi}_{\mathrm{adv}}\circ\boldsymbol{\pi}\circ\boldsymbol{f}_{\mathrm{adv}}}\left[r_{t+1} + \gamma V_{\boldsymbol{\pi}_{\mathrm{adv}}\circ\boldsymbol{\pi}\circ\boldsymbol{f}_{\mathrm{adv}}}(s_{t+1}) \,\big|\, s_t\right] \\
&= \mathbb{E}_{\tilde{\boldsymbol{a}}_t \sim \boldsymbol{\pi}_{\mathrm{adv}}\circ\boldsymbol{\pi}\circ\boldsymbol{f}_{\mathrm{adv}}}\left[\mathbb{E}_{s_{t+1}\sim P(\cdot|s_t,\tilde{\boldsymbol{a}}_t)}\big[R(s_t,\tilde{\boldsymbol{a}}_t,s_{t+1}) + \gamma V_{\boldsymbol{\pi}_{\mathrm{adv}}\circ\boldsymbol{\pi}\circ\boldsymbol{f}_{\mathrm{adv}}}(s_{t+1})\big]\right].
\end{aligned}$$

Now, we prove Theorem 4.2 based on Lemma A.1.

**Theorem 4.2.** *Given a JA-Dec-POMDP $\mathcal{M}^J$ with joint attackers $\boldsymbol{f}_{\mathrm{adv}}$ and $\boldsymbol{\pi}_{\mathrm{adv}}$, there exists an induced Dec-POMDP $\tilde{\mathcal{M}} = \langle \mathcal{N}, \tilde{\mathcal{S}}, \mathcal{A}, \tilde{P}, \Omega, \tilde{O}, \tilde{R}, \gamma \rangle$, whose state space, transition probability, observation, and reward functions explicitly account for the joint attack. Specifically,*

$$\begin{aligned}
\tilde{\boldsymbol{o}}_t &:= \tilde{O}(s_t, i) = \boldsymbol{f}_{\mathrm{adv}}(\cdot \mid O(s_t, i)), \quad \hat{\boldsymbol{a}}_t \sim \boldsymbol{\pi}(\cdot|\tilde{\boldsymbol{o}}_t), \\
\tilde{P}(\tilde{s}_{t+1} \mid \tilde{s}_t, \hat{\boldsymbol{a}}_t) &:= P(s_{t+1} \mid s_t, \hat{\boldsymbol{a}}_t) \cdot \boldsymbol{\pi}_{\mathrm{adv}}(\tilde{\boldsymbol{a}}_t \mid s_t, \hat{\boldsymbol{a}}_t), \\
\tilde{R}(\tilde{s}_t, \hat{\boldsymbol{a}}_t, \tilde{s}_{t+1}) &:= R(s_t, \tilde{\boldsymbol{a}}_t, s_{t+1}),
\end{aligned}$$

*where $\tilde{s}_t := (s_t, \tilde{\boldsymbol{a}}_t) \in \tilde{\mathcal{S}}$ with $\tilde{\boldsymbol{a}}_t \sim \boldsymbol{\pi}_{\mathrm{adv}} \circ \boldsymbol{\pi} \circ \boldsymbol{f}_{\mathrm{adv}}$. Then, for any joint policy $\boldsymbol{\pi}$ and all $s_t \in \mathcal{S}$, the state value $\tilde{V}_{\boldsymbol{\pi}}(s_t)$ of $\boldsymbol{\pi}$ in the Dec-POMDP $\tilde{\mathcal{M}}$ is equivalent to the state value $V_{\boldsymbol{\pi}_{\mathrm{adv}}\circ\boldsymbol{\pi}\circ\boldsymbol{f}_{\mathrm{adv}}}(s_t)$ in the JA-Dec-POMDP $\mathcal{M}^J$.*

*Proof.* From the defined perturbed state space, transition probability, observation, and reward functions, we can express the state-value function $\tilde{V}_{\boldsymbol{\pi}}$ of the induced Dec-POMDP $\tilde{\mathcal{M}}$ in Bellman form as follows.

$$\begin{aligned}
\tilde{V}_{\boldsymbol{\pi}}(s_t) := \tilde{V}_{\boldsymbol{\pi}}(\tilde{s}_t) &= \mathbb{E}_{\hat{\boldsymbol{a}}_t \sim \boldsymbol{\pi}(\cdot|\tilde{O}(s_t))}\left[\mathbb{E}_{\tilde{s}_{t+1}\sim \tilde{P}(\tilde{s}_{t+1}|\tilde{s}_t,\hat{\boldsymbol{a}}_t)}[\tilde{R}(\tilde{s}_t,\hat{\boldsymbol{a}}_t,\tilde{s}_{t+1}) + \gamma \tilde{V}_{\boldsymbol{\pi}}(s_{t+1})]\right] \\
&\overset{(a)}{=} \mathbb{E}_{\tilde{\boldsymbol{o}}_t \sim \boldsymbol{f}_{\mathrm{adv}}(\tilde{\boldsymbol{o}}_t|O(s_t)),\, \hat{\boldsymbol{a}}_t \sim \boldsymbol{\pi}(\cdot|\tilde{\boldsymbol{o}}_t)}\left[\mathbb{E}_{\tilde{\boldsymbol{a}}_t \sim \boldsymbol{\pi}_{\mathrm{adv}}(\cdot|s_t,\hat{\boldsymbol{a}}_t),\, s_{t+1}\sim P(\cdot|s_t,\hat{\boldsymbol{a}}_t)}[R(s_t,\tilde{\boldsymbol{a}}_t,s_{t+1}) + \gamma \tilde{V}_{\boldsymbol{\pi}}(s_{t+1})]\right] \\
&= \mathbb{E}_{\tilde{\boldsymbol{a}}_t \sim \boldsymbol{\pi}_{\mathrm{adv}}\circ\boldsymbol{\pi}\circ\boldsymbol{f}_{\mathrm{adv}}}\left[\mathbb{E}_{s_{t+1}\sim P(\cdot|s_t,\tilde{\boldsymbol{a}}_t)}\big[R(s_t,\tilde{\boldsymbol{a}}_t,s_{t+1}) + \gamma \tilde{V}_{\boldsymbol{\pi}}(s_{t+1})\big]\right],
\end{aligned} \tag{4}$$

where (a) follows directly from the definitions of the perturbed components in the induced Dec-POMDP.

From Lemma A.1, the Bellman equations for $\tilde{V}_{\boldsymbol{\pi}}$ and $V_{\boldsymbol{\pi}_{\mathrm{adv}}\circ\boldsymbol{\pi}\circ\boldsymbol{f}_{\mathrm{adv}}}$ coincide exactly. By the fixed-point theorem, the value function satisfying the Bellman equation is unique; hence, the two values are equivalent.

## A.2. Proof of Lemma 4.3

**Lemma 4.3.** *Given a masking attacker $\boldsymbol{m}^{G_1}$, the group-wise observation-level MI can be upper-bounded as*

$$
\begin{aligned}
& \mathcal{I}(\tilde{\boldsymbol{o}}_{t+1}^{G_1}; \boldsymbol{a}_t^{G_2} \mid \boldsymbol{a}_t^{G_1}, \boldsymbol{\tau}_t) \\
& \leq \sum_{i \in G_1} \sum_{d \notin D^i} \underbrace{\sum_{j \in G_2} \mathcal{I}(o_{d,t+1}^i; a_t^j \mid a_t^i, \boldsymbol{\tau}_t)}_{\text{dimension-wise MI}} + \mathcal{R}(G_1; G_2),
\end{aligned}
\tag{5}
$$

*where $d_o$ denotes the observation dimension and $\mathcal{R}(G_1; G_2)$ denotes a group redundancy term.*

*Proof.* It is known that multivariate mutual information admits the following Möbius expansion (Matsuda, 2000; Singha & Shenoy, 2018). Let $\boldsymbol{X} = (X_1, \ldots, X_n)$ be a multivariate vector, and let $Y$ and $Z$ denote random variables. Then the mutual information $\mathcal{I}(\boldsymbol{X}; Y \mid Z)$ can be expanded as

$$
\begin{aligned}
\mathcal{I}(\boldsymbol{X}; Y \mid Z) &= \sum_{i=1}^n \mathcal{I}(X_i; Y \mid Z) - \sum_{i<j} \mathcal{I}(X_i; X_j; Y \mid Z) + \cdots + (-1)^{n+1} \mathcal{I}(X_1; \ldots; X_n; Y \mid Z) \\
&= \sum_{i=1}^n \mathcal{I}(X_i; Y \mid Z) + \mathcal{R}(\boldsymbol{X}; Y \mid Z),
\end{aligned}
\tag{6}
$$

where $\mathcal{I}(X_i; X_j; Y \mid Z) = \mathcal{I}(X_i; X_j \mid Z) - \mathcal{I}(X_i; X_j \mid Y, Z)$ denotes the *interaction information* among $X_i$, $X_j$, $Y$ given $Z$, and $\mathcal{R}(\boldsymbol{X}; Y \mid Z)$ is the redundancy term.

Then, the group-wise MI $\mathcal{I}(\tilde{\boldsymbol{o}}_{t+1}^{G_1}; \boldsymbol{a}_t^{G_2} \mid \boldsymbol{a}_t^{G_1}, \boldsymbol{\tau}_t)$ can be written as

$$
\begin{aligned}
\mathcal{I}(\tilde{\boldsymbol{o}}_{t+1}^{G_1}; \boldsymbol{a}_t^{G_2} \mid \boldsymbol{a}_t^{G_1}, \boldsymbol{\tau}_t) &\underset{(a)}{=} \sum_{i \in G_1} \sum_{j \in G_2} \sum_{d=1}^{d_o} \mathcal{I}(\tilde{o}_{d,t+1}^i; a_t^j \mid \boldsymbol{a}^{G_1}, \boldsymbol{\tau}_t) + \mathcal{R}(G_1; G_2) \\
&\underset{(b)}{\leq} \sum_{i \in G_1} \sum_{j \in G_2} \sum_{d=1}^{d_o} \mathcal{I}(\tilde{o}_{d,t+1}^i; a_t^j \mid a_t^i, \boldsymbol{\tau}_t) + \mathcal{R}(G_1; G_2) \\
&= \sum_{i \in G_1} \sum_{j \in G_2} \left( \sum_{d \notin D^i} \mathcal{I}(\tilde{o}_{d,t+1}^i; a_t^j \mid a_t^i, \boldsymbol{\tau}_t) + \sum_{d \in D^i} \mathcal{I}(\tilde{o}_{d,t+1}^i; a_t^j \mid a_t^i, \boldsymbol{\tau}_t) \right) + \mathcal{R}(G_1; G_2) \\
&\underset{(c)}{=} \sum_{i \in G_1} \sum_{j \in G_2} \left( \sum_{d \notin D^i} \mathcal{I}(o_{d,t+1}^i; a_t^j \mid a_t^i, \boldsymbol{\tau}_t) + \sum_{d \in D^i} \overset{0}{\cancel{\mathcal{I}(0; a_t^j \mid a_t^i, \boldsymbol{\tau}_t)}} \right) + \mathcal{R}(G_1; G_2) \\
&= \sum_{i \in G_1} \sum_{d \notin D^i} \underbrace{\sum_{j \in G_2} \mathcal{I}(o_{d,t+1}^i; a_t^j \mid a_t^i, \boldsymbol{\tau}_t)}_{\text{dimension-wise MI}} + \mathcal{R}(G_1; G_2).
\end{aligned}
\tag{7}
$$

In (7), (a) follows directly by applying the Möbius expansion (6) sequentially, decomposing across agents in $G_1$ and $G_2$ and across all observation dimensions in $G_1$, where the group redundancy term $\mathcal{R}(G_1; G_2)$ is defined as the sum of all terms that arise in the expansion except the dimension-wise MI terms. (b) follows from the conditioning relaxation property of mutual information, i.e., $\mathcal{I}(X; Y \mid Z_1, Z_2) \leq \mathcal{I}(X; Y \mid Z_1)$. Finally, for (c), given the masking dimensions $D^i$, we have $\tilde{o}_{d,t+1}^i = o_{d,t+1}^i$ for $d \notin D^i$ (unmasked dimensions) and $\tilde{o}_{d,t+1}^i = 0$ for $d \in D^i$ (masked dimensions). Since $0$ is a deterministic constant, the corresponding mutual information satisfies $\mathcal{I}(0; a_t^j \mid a_t^i, \boldsymbol{\tau}_t) = 0$.

## B. Implementation Details

We provide additional implementation details for our attacks and training pipeline. Section B.1 describes the MI estimation required for MI-based observation and action attacks. Section B.2 presents the MARL implementation used to train IBAL under the Interaction-Breaking attack.

### B.1. Mutual Information Estimation

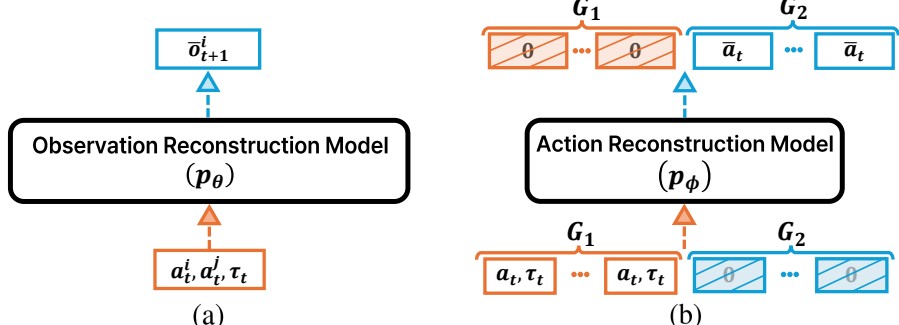

*Figure 10.* Reconstruction models for MI estimation: (a) observation reconstruction model and (b) action reconstruction model.

The proposed joint interaction-breaking attackers $f_{\mathrm{adv}}^{\mathrm{IB}}$ and $\pi_{\mathrm{adv}}^{\mathrm{IB}}$ requires estimating the associated MI terms. In particular, Eq. (1) decomposes the total interaction into the action-level MI term and the observation-level MI term, both of which must be estimated to construct the attacker. To this end, we build two MI estimators: one for the observation-level MI and the other for the action-level MI.

**Observation-level MI Estimation.** Following Definition 4.4, this requires estimating $\mathcal{I}(o_{t+1}^i; a_t^j \mid a_t^i, \boldsymbol{\tau}_t)$. However, directly computing this is generally intractable (Paninski, 2003). Since our attacker aims to reduce this MI, we adopt the CLUB (Cheng et al., 2020) upper bound and use its sample-based estimate as a surrogate for the observation-level MI.

Following CLUB, we upper-bound

$$\mathcal{I}\left(o_{t+1}^i; a_t^j \mid a_t^i, \boldsymbol{\tau}_t\right) \ \leq \ \mathcal{I}_{\mathrm{CLUB}}\left(o_{t+1}^i; a_t^j \mid a_t^i, \boldsymbol{\tau}_t\right).$$

We estimate $\mathcal{I}_{\mathrm{CLUB}}$ from samples. Let $\mathcal{D}^+$ be a positive buffer containing aligned tuples $\{(o_{t+1}^i, a_t^i, a_t^j, \boldsymbol{\tau}_t)\}$ drawn from the same transition. We construct negative pairs by replacing $a_t^j$ with an action sampled independently from a negative buffer $\mathcal{D}^-$ (e.g., sampling $a_t^j$ from other transitions) to break the dependence between $o_{t+1}^i$ and $a_t^j$. Given a batch $\{(o_{t+1}^i, a_t^i, a_t^j, \boldsymbol{\tau}_t)_b)\}_{b=1}^B \sim \mathcal{D}^+$ and negative indices $k_b$ uniformly sampled from $\{1, \ldots, B\}$, we approximate the CLUB:

$$\mathcal{I}_{\mathrm{CLUB}}\left(o_{t+1}^i; a_t^j \mid a_t^i, \boldsymbol{\tau}_t\right) \approx \frac{1}{B} \sum_{b=1}^B \log p_\theta\Big((o_{t+1}^i)_b \,\Big|\, (a_t^i, a_t^j, \boldsymbol{\tau}_t)_b\Big)$$

$$- \frac{1}{B} \sum_{b=1}^B \log p_\theta\Big((o_{t+1}^i)_b \,\Big|\, (a_t^j)_{k_b}, (a_t^i, \boldsymbol{\tau}_t)_b\Big).$$

where $p_\theta$ is an agent-wise observation reconstruction model (Fig. 10(a)) that predicts $o_{t+1}^i$ from $(a_t^i, a_t^j, \boldsymbol{\tau}_t)$. We parameterize $p_\theta(o_{t+1}^i \mid a_t^i, a_t^j, \boldsymbol{\tau}_t)$ as a diagonal Gaussian with mean $\mu_\theta$ and log-variance $\log v_\theta$, where the reconstructed observation is $\bar{o}_{t+1}^i := \mu_\theta(a_t^i, a_t^j, \boldsymbol{\tau}_t)$. We train the model by minimizing the negative log-likelihood:

$$\mathcal{L}(\theta) = \mathbb{E}\Big[ -\log p_\theta\Big(o_{t+1}^i \mid a_t^i, a_t^j, \boldsymbol{\tau}_t\Big) \Big].$$

In practice, we do not estimate MI at every timestep. Instead, we periodically collect a batch of samples and compute the average agent-wise MI, which are then used by the observation attacker.

**Action-level MI Estimation.** Following Definition 4.5, we estimate the action-level MI term $\mathcal{I}\big(\boldsymbol{a}_t^{G_1}; \boldsymbol{a}_t^{G_2} \mid \boldsymbol{\tau}_t\big)$. This conditional MI admits a closed-form KL characterization:

$$\mathcal{I}\big(\boldsymbol{a}_t^{G_1}; \boldsymbol{a}_t^{G_2} \mid \boldsymbol{\tau}_t\big) = D_{\mathrm{KL}}\Big(p(\boldsymbol{a}_t^{G_2} \mid \boldsymbol{a}_t^{G_1}, \boldsymbol{\tau}_t) \,\Big\|\, p(\boldsymbol{a}_t^{G_2} \mid \boldsymbol{\tau}_t)\Big),$$

which requires estimating the conditional distributions $p(\boldsymbol{a}_t^{G_2} \mid \boldsymbol{a}_t^{G_1}, \boldsymbol{\tau}_t)$ and $p(\boldsymbol{a}_t^{G_2} \mid \boldsymbol{\tau}_t)$. We approximate these with an action reconstruction model $p_\phi(\boldsymbol{a}_t^{G_2} \mid \boldsymbol{a}_t^{G_1}, \boldsymbol{\tau}_t)$. To avoid training separate models for each $(G_1, G_2)$ pair, we train a single model $p_\phi$ over the joint action $\boldsymbol{a}_t$ (Jaques et al., 2019), whose overall design is illustrated in Fig. 10(b). Concretely, we zero-mask the $G_2$ components in the input action and train $p_\phi$ to reconstruct the $G_2$ actions from $(\boldsymbol{a}_t^{G_1}, \boldsymbol{\tau}_t)$. For training, we form the target by zero-masking the $G_1$ components in the output action so that the loss is computed only on $G_2$. Let $\bar{\boldsymbol{a}}_t$ denote the reconstructed action, whose $G_2$ components $\bar{\boldsymbol{a}}_t^{G_2}$ are used as the reconstruction outputs. Accordingly, we minimize the cross-entropy loss

$$\mathcal{L}(\phi) = \mathbb{E}\Big[-\sum_{j \in G_2} \log p_\phi\big(a_t^j \mid \boldsymbol{a}_t^{G_1}, \boldsymbol{\tau}_t\big)\Big].$$

In practice, for each attacked agent $i \in G_1$, we counterfactually evaluate available actions and choose the one that minimizes our estimated action-level MI objective. In addition, Appendix F.4 shows the computational complexity comparison; despite the additional MI estimation, the overall training time increases only slightly compared to Vanilla QMIX.

## B.2. Detailed Implementation of IBAL Framework

With the interaction-breaking attackers, we obtain the JA-Dec-POMDP $\mathcal{M}^J$ and its induced Dec-POMDP $\tilde{\mathcal{M}}$. As summarized by Theorem 4.2, IBAL trains CTDE-based MARL policies on $\tilde{\mathcal{M}}$ to improve robustness under interaction breaking in $\mathcal{M}^J$. In our implementation, we instantiate IBAL with the representative value-based CTDE algorithm **QMIX** (Rashid et al., 2020b) by applying the interaction-breaking adversarial attacks.

QMIX learns decentralized policies by training individual agent $Q$-networks and combining them with a monotonic mixing network to estimate the joint action-value $Q^{\mathrm{tot}}$ under centralized training. IBAL follows the same $Q$-learning procedure as QMIX, but collects replay data from $\tilde{\mathcal{M}}$, where interaction-breaking perturbations are applied. We minimize the temporal-difference (TD) loss on the joint value:

$$\mathcal{L}_{\mathrm{TD}}(\psi) = \mathbb{E}\Big[\big(r_t + \gamma \max_{\boldsymbol{a}'} Q_{\psi^{\mathrm{targ}}}^{\mathrm{tot}}(s_{t+1}, \boldsymbol{a}') - Q_\psi^{\mathrm{tot}}(s_t, \boldsymbol{a}_t)\big)^2\Big],$$

where $\psi^{\mathrm{targ}}$ denotes the target-network parameters updated via an exponential moving average of $\psi$.

## C. Environmental Details

This section provides detailed descriptions of the experimental environments used in our experiments, including SMAC (Samvelyan et al., 2019) and SMACv2 (Ellis et al., 2023) in Appendix C.1 and Level-Based Foraging (LBF) (Papoudakis et al., 2020) in Appendix C.2.

### C.1. The StarCraft Multi-Agent Challenge (SMAC & SMACv2)

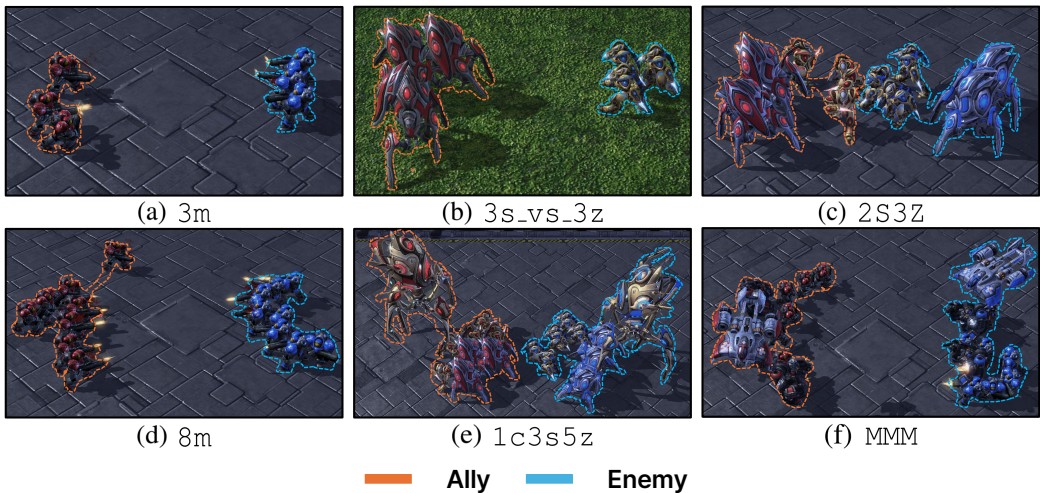

(a) 3m    (b) 3s_vs_3z    (c) 2S3Z

(d) 8m    (e) 1c3s5z    (f) MMM

━━ Ally    ━━ Enemy

*Figure 11.* Visualization of SMAC scenarios.

SMAC is a standard testbed for cooperative multi-agent reinforcement learning, designed around decentralized micro-management in StarCraft II. In SMAC, each unit is controlled by an individual agent that makes decisions from its own partial, local observations without access to global state at execution time. The benchmark provides diverse combat settings, enabling systematic evaluation of coordination and robustness across scenarios.

We additionally consider SMACv2, which extends SMAC by introducing procedural variation into the scenarios. While SMAC uses fixed unit compositions and initial positions for each map, SMACv2 randomizes team compositions and spawn positions across episodes. This makes SMACv2 more stochastic and challenging, as agents must adapt their coordination strategies to varying battlefield configurations rather than relying on a fixed scenario structure.

**Scenarios.** SMAC scenarios simulate combat engagements between an allied squad controlled by learning agents and an opposing squad driven by a scripted AI. Tasks vary in unit matchups, map layouts, and terrain structure, requiring micromanagement skills such as focus fire, kiting, and positional play around choke points and obstacles. An episode terminates when either side's units are eliminated or when the time limit is reached, and the learning objective is to maximize the allied team's expected return. Scenario specifications and unit compositions are summarized in Table 2 and Fig. 11.

**State and Observation Spaces.** SMAC follows a centralized-training decentralized-execution (CTDE) protocol. At execution time, each agent receives a local observation under partial observability: it only reflects entities within a fixed sight range of 9 units and excludes global information. Concretely, the observation is composed of (i) movement-related features, (ii) per-enemy features such as relative position, distance, and combat-relevant attributes (e.g., health, shield, and unit type), (iii) per-ally features of nearby teammates in a similar form to per-enemy features, and (iv) the agent's own features. During centralized training, a global state is additionally available, which aggregates features of all allied and enemy units together with the last actions. The precise state and observation dimensions depend on the map and are summarized in Table 2.

**Action Space.** Agents operate with a discrete action set that includes movement in the four directions (North, South, East, West), attacking an enemy, and unit-specific abilities such as healing (e.g., Medivacs). In addition, each agent can issue a *stop* command or take a *no-op* action, where *no-op* is only available for dead units. The action-space size depends on the scenario and is given by $n_{\text{actions}} = 6 + n_{\text{enemies}}$, where the constant 6 accounts for movement actions together with stop and no-op, and the $n_{\text{enemies}}$ term corresponds to selecting which enemy to target when executing an attack. Since $n_{\text{enemies}}$ varies across maps, the action-space size differs by scenario, as summarized in Table 2.

**Reward Function.** SMAC provides a shaped team reward, defined on each transition as $r_t = R(s_t, \boldsymbol{a}_t, s_{t+1})$. It combines (i) damage dealt to enemies, (ii) a bonus for enemy kills, and (iii) a terminal win bonus:

$$r_t = \sum_{e \in \mathcal{E}} \Big(\mathrm{Health}_t(e) - \mathrm{Health}_{t+1}(e)\Big) + \mathrm{Reward}_{\mathrm{kill}} \sum_{e \in \mathcal{E}} \mathbb{I}[\mathrm{Health}_{t+1}(e) = 0] + \mathrm{Reward}_{\mathrm{win}} \cdot \mathbb{I}[\mathrm{win}_{t+1} = \mathrm{True}],$$

where $\mathcal{E}$ is the set of enemy units and $\mathrm{Health}_t(e)$ denotes the health of enemy $e$ at time $t$. The indicator function $\mathbb{I}[\cdot]$ equals 1 if the condition holds and 0 otherwise. We use the standard SMAC scaling with $\mathrm{Reward}_{\mathrm{kill}} = 10$ and $\mathrm{Reward}_{\mathrm{win}} = 200$.

*Table 2.* Configuration for each SMAC and SMACv2 scenario.

| Map | Ally Units | Enemy Units | State Dimension | Obs. Dimension | Num. of Actions |
|---|---|---|---|---|---|
| 3m | 3 Marines | 3 Marines | 48 | 30 | 9 |
| 3s_vs_3z | 3 Stalkers | 3 Zealots | 54 | 36 | 9 |
| 2s3z | 2 Stalkers, 3 Zealots | 2 Stalkers, 3 Zealots | 120 | 80 | 11 |
| 8m | 8 Marines | 8 Marines | 168 | 80 | 14 |
| 1c3s5z | 1 Colossus, 3 Stalkers, 5 Zealots | 1 Colossus, 3 Stalkers, 5 Zealots | 270 | 162 | 15 |
| MMM | 1 Medivac, 2 Marauders, 7 Marines | 1 Medivac, 2 Marauders, 7 Marines | 290 | 160 | 16 |
| terran_5_vs_5 | 5 Terran units | 5 Terran units | 65 | 82 | 11 |
| zerg_5_vs_5 | 5 Zerg units | 5 Zerg units | 65 | 82 | 11 |
| protoss_5_vs_5 | 5 Protoss units | 5 Protoss units | 75 | 92 | 11 |

## C.2. Level-Based Foraging (LBF)

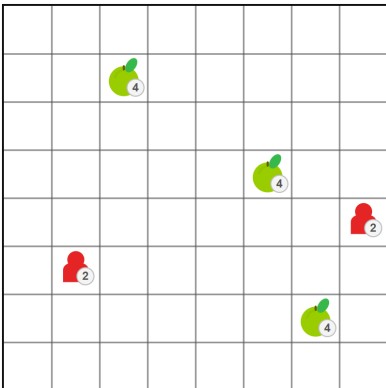

*Figure 12.* Visualization of LBF.

Level-Based Foraging (LBF) is a cooperative multi-agent grid-world benchmark in which agents collect food items distributed across the map. Each agent and food item has an associated level, and a food item can be collected only when the total level of the adjacent agents attempting to load it is at least as large as the food level. This structure requires agents to coordinate which food to approach and when to execute the loading action.

**Scenarios.** We use LBF scenarios of the form `Foraging-20x20-{5p,10p,15p}-6f-s3`, where agents operate on a $20 \times 20$ grid with 6 food items and a sight range of 3. The number of agents varies across scenarios, allowing us to evaluate IBAL under different group sizes and coordination demands. An episode terminates when all food items are collected or when the time limit is reached.

**State and Observation Spaces.** Each agent receives a local observation restricted to entities within its sight range, including nearby agents, nearby food items, their positions, and their levels. Under CTDE, a global state containing information about all agents and food items is available during centralized training, while each agent acts only from its local observation during decentralized execution.

**Action Space.** Agents use a discrete action space consisting of movement actions, a *load* action for collecting adjacent food, and a *no-op* action. Successful collection requires sufficient joint level from the agents participating in the loading action, making coordination necessary when high-level food cannot be collected by a single agent.

**Reward Function.** LBF provides a cooperative team reward when food is successfully collected. The reward depends on the collected food item and is shared among agents in the cooperative setting. Thus, agents must learn to coordinate their movement and loading decisions to maximize the total collected reward within the episode.

# D. Experimental Details

All experiments are run on a dedicated GPU machine with an NVIDIA GeForce RTX 3090 and an AMD EPYC 7513 (32 cores), using Ubuntu 20.04 and PyTorch. Hyperparameter configurations are detailed in Appendix D.1, and additional descriptions of adversarial attack methods and robust MARL baselines are provided in Appendix D.2.

## D.1. Hyperparameter Setup

All CTDE baselines use the default hyperparameter configuration from PyMARL (Samvelyan et al., 2019), and the shared CTDE settings are summarized in Table 3. We only tune the additional hyperparameters introduced by the interaction-breaking attack and IBAL via parameter search; the selected values are reported in Table 4.

Across all SMAC scenarios, we fix two parameters in the adaptive attack probability schedule: the growth rate $\alpha = 1.1$, which controls how aggressively the attack probability is increased, and the success-rate threshold $\eta = 0.8$, which determines when the probability is updated. These values provided the most stable training behavior in our experiments. Task-specific hyperparameters include the maximum group size $K$, which specifies the maximum number of agents in the group $G_1$ and thereby governs the diversity of the sampled $(G_1, G_2)$ partitions, and the masking budget $L$, defined as the per-agent size of the masked index set $\{D^i\}_{i \in G_1}$ in the MI-based observation attack, which controls the perturbation strength; we set $L \propto |G_2|$. We also tune the minimum attack probability, which lower-bounds the action attack intensity; we set the minimum attack probability to $1/K$. For evaluation, we fix the action attack probability to the minimum attack probability $1/K$ as well. All task-specific hyperparameters are selected based on our ablation study, which is provided in Appendix F.3.

*Table 3.* Common hyperparameters of IBAL.

| Hyperparameter | Value |
| --- | --- |
| Epsilon | $1.0 \rightarrow 0.05$ |
| Epsilon anneal time | 50000 timesteps |
| Gamma | 0.99 |
| Buffer size | 5000 episodes |
| Batch size | 32 episodes |
| Critic loss | MSE Loss |
| Critic learning rate | 0.0005 |
| Optimizer | RMSProp |
| Growth rate ($\alpha$) | 1.1 |
| Success rate threshold ($\eta$) | 0.8 |

*Table 4.* Task-specific hyperparameters of IBAL on SMAC tasks.

| Scenario | 3m | 3s_vs_3z | 2s3z | 8m | 1c3s5z | MMM |
| --- | --- | --- | --- | --- | --- | --- |
| Maximum group size ($K$) | 1 | 1 | 1 | 4 | 4 | 4 |
| Masking budget ($L/|G_2|$) | 5 | 5 | 8 | 5 | 8 | 8 |

*Table 5.* Task-specific hyperparameters of IBAL on SMACv2 tasks.

| Scenario | Terran 5 vs 5 | Zerg 5 vs 5 | Protoss 5 vs 5 |
| --- | --- | --- | --- |
| Maximum group size ($K$) | 1 | 1 | 1 |
| Masking budget ($L/|G_2|$) | 8 | 8 | 8 |

*Table 6.* Task-specific hyperparameters of IBAL on LBF tasks (Foraging-20x20-$n$p-6f-s3).

| Scenario | 5p | 10p | 15p |
| --- | --- | --- | --- |
| Maximum group size ($K$) | 2 | 5 | 7 |
| Masking budget ($L/|G_2|$) | 3 | 3 | 3 |

## D.2. Detailed Description for Other Baselines

We consider several adversarial perturbation schemes and the corresponding robust baselines trained against them. Unless otherwise stated, each robust baseline is implemented by training QMIX under the associated attack within the PyMARL framework (https://github.com/oxwhirl/pymarl).

**Natural (Nat.) & Vanilla QMIX**: The no-attack setting where agents act on clean observations and execute their sampled actions without perturbations; Vanilla QMIX denotes the QMIX policy trained under this natural setting.

**Rand. & Rand-Obs / Rand-Act**: In the random observation attack, at each timestep we select an agent $i$ uniformly at random and inject Gaussian noise to its observation,

$$\tilde{o}_t^i = o_t^i + \epsilon_t, \quad \epsilon_t \sim \mathcal{N}(\mathbf{0}, \sigma^2 \mathbf{I}), \ \sigma = 0.1.$$

while in the random action attack, with probability $30\%$ we sample one agent $i$ uniformly and force a random legal action,

$$\tilde{a}_t^i \sim \mathrm{Unif}(\mathcal{A}_t^i).$$

Rand. denotes the combined setting where both perturbations are applied simultaneously, and Rand-Obs and Rand-Act denote QMIX policies trained under the random observation and random action attacks, respectively.

**FGSM** (Goodfellow et al., 2014): FGSM is a gradient-based observation attack. At each timestep, we sample an agent $i$ uniformly and perturb its observation in the direction that reduces its action-value $Q^i$,

$$\tilde{o}_t^i = o_t^i - \epsilon \cdot \mathrm{sign}\Big(\nabla_{o_t^i} Q^i(\tau_t^i, a_t^i)\Big), \quad \epsilon = 0.1.$$

FGSM (robust baseline) denotes QMIX trained under this FGSM observation attack.

**ATLA** (Zhang et al., 2021a): ATLA trains an RL-based observation attacker that generates bounded perturbations to minimize the team return. Following ATLA, we learn a PPO-based attacker that outputs an additive noise $\epsilon_t^i$ and applies it to a randomly chosen agent $i$:

$$\tilde{o}_t^i = o_t^i + \epsilon_t^i.$$

The attacker is trained with PPO using the negative reward $(-r_t)$. ATLA (robust baseline) denotes QMIX trained against the learned ATLA observation attacker.

**ERNIE** (Bukharin et al., 2023): ERNIE improves robustness via adversarial regularization that encourages Lipschitz-like behavior of each agent $k$'s policy under bounded observation perturbations:

$$R_{\boldsymbol{\pi}}(o_k; \theta_k) = \max_{\|\delta\| \le \epsilon} D\big(\pi_{\theta_k}(o_k + \delta), \pi_{\theta_k}(o_k)\big),$$

where $D$ is a divergence measure (e.g., KL-divergence). ERNIE casts robust training in a Stackelberg-game form and provides scalable variants such as mean-field MARL. We evaluate using its official implementation: https://github.com/abukharin3/ERNIE.

**ROMANCE** (Yuan et al., 2023): ROMANCE improves robustness by generating a diverse set of auxiliary adversarial attackers through an evolutionary mechanism. It optimizes a combined objective that trades off attack strength and diversity:

$$L_{\mathrm{adv}}(\phi) = \frac{1}{n_p} \sum_{j=1}^{n_p} L_{\mathrm{opt}}(\phi_j) - \alpha L_{\mathrm{div}}(\phi),$$

where $L_{\mathrm{opt}}$ reduces the ego-system return and $L_{\mathrm{div}}$ promotes diversity via Jensen–Shannon divergence. We evaluate using its official implementation: https://github.com/zzq-bot/ROMANCE.

**WALL** (Lee et al., 2025): Wolfpack Attack is a coordinated action-perturbation framework inspired by wolf hunting strategies: it first targets an initial agent, then selects follow-up agents (e.g., those exhibiting the largest policy change) to amplify disruption, and uses a model-based procedure to identify timesteps where the attack is most critical. WALL denotes a robust QMIX policy trained against Wolfpack Attack. We evaluate using its official implementation: https://github.com/sunwoolee0504/WALL.

# E. Additional Experiments

This section presents full results of performance comparisons. Appendix E.1 reports full performance tables with standard deviations under adversarial attacks and provides learning curves evaluated against an unseen interaction-breaking attack over training timesteps. Appendix E.2 covers full performance tables with standard deviations under non-parametric perturbations. Appendix E.3 further provides additional performance comparison results on LBF and SMACv2. Appendix E.4 reports additional comparisons with other CTDE algorithms, particularly MAPPO.

## E.1. Full Performance Comparison Result under Adversarial Attack

Table 7, 8 reports full results with standard deviations under adversarial attacks, where IBAL achieves consistently higher win rates and generalizes better across observation and action attacks. While baselines such as FGSM and WALL often perform well on their own attacks, their performance drops sharply under the proposed interaction-breaking attack, whereas IBAL remains robust. Fig. 13 reports training learning curves, where each method is trained under its own standard setup and periodically evaluated under an unseen interaction-breaking attack. Most baselines show little improvement under this evaluation despite learning in their respective training regimes, indicating that robustness does not carry over when cooperation collapses, whereas IBAL steadily learns to counteract the disruption.

*Table 7.* Success rates (%) under various adversarial attacks on SMAC scenarios: 3m, 3s_vs_3z, and 2s3z. **Bold** indicates the best and underlining indicates the second-best.

| Scenario | Model | Attack | | | | | | Mean |
|---|---|---|---|---|---|---|---|---|
| | | Nat. | Rand. | FGSM | EGA | Wolf. | Ours | |
| 3m | Vanilla QMIX | 96.0 ± 3.12 | 45.0 ± 7.00 | 76.3 ± 5.86 | 62.5 ± 14.3 | 38.8 ± 8.46 | 0.00 ± 0.00 | 53.1 ± 3.20 |
| | Rand-Obs | 96.9 ± 2.64 | 48.7 ± 4.93 | 86.0 ± 0.94 | 54.2 ± 19.1 | 44.8 ± 4.77 | 0.00 ± 0.00 | 55.4 ± 3.41 |
| | Rand-Act | 96.8 ± 2.95 | 42.3 ± 0.58 | 79.7 ± 8.08 | 83.3 ± 3.61 | 43.8 ± 3.12 | 0.00 ± 0.00 | 57.6 ± 1.64 |
| | FGSM | 94.7 ± 0.80 | 56.7 ± 2.52 | **96.3 ± 2.08** | 45.8 ± 11.0 | 46.9 ± 11.3 | 1.00 ± 1.80 | 56.9 ± 2.70 |
| | ATLA | 97.2 ± 1.54 | 58.0 ± 1.00 | 86.3 ± 4.04 | 45.8 ± 6.51 | 45.6 ± 1.66 | 0.00 ± 0.00 | 55.5 ± 1.34 |
| | ERNIE | 96.9 ± 1.27 | 46.3 ± 9.24 | 85.0 ± 11.1 | 41.7 ± 9.55 | 36.5 ± 9.02 | 5.20 ± 9.02 | 51.9 ± 3.59 |
| | ROMANCE | 97.1 ± 2.89 | 49.7 ± 2.08 | 72.0 ± 4.58 | 88.5 ± 3.61 | 41.0 ± 13.1 | 1.00 ± 1.80 | 58.2 ± 2.49 |
| | WALL | 97.2 ± 3.06 | 48.3 ± 6.81 | 70.0 ± 13.0 | 86.7 ± 10.2 | **83.3 ± 7.87** | 0.00 ± 0.00 | 64.2 ± 3.29 |
| | IBAL (ours) | **98.6 ± 0.97** | **62.3 ± 4.51** | 87.1 ± 4.89 | **89.0 ± 1.44** | 70.1 ± 7.88 | 78.3 ± 1.84 | **80.9 ± 1.77** |
| 3s_vs_3z | Vanilla QMIX | 99.2 ± 0.55 | 61.2 ± 3.40 | 67.8 ± 18.1 | 71.9 ± 7.21 | 66.7 ± 7.86 | 6.20 ± 5.41 | 62.2 ± 3.91 |
| | Rand-Obs | 98.4 ± 0.78 | 86.0 ± 5.29 | 78.0 ± 3.46 | 87.1 ± 5.74 | 89.0 ± 2.53 | 11.4 ± 6.53 | 75.0 ± 1.64 |
| | Rand-Act | 97.5 ± 1.72 | 76.0 ± 3.46 | 61.7 ± 11.6 | 75.4 ± 5.77 | 91.7 ± 1.80 | 10.4 ± 3.61 | 68.0 ± 2.69 |
| | FGSM | 98.5 ± 1.50 | 87.3 ± 0.58 | **98.8 ± 0.72** | 68.8 ± 16.2 | 90.6 ± 5.41 | 24.0 ± 3.61 | 78.7 ± 2.91 |
| | ATLA | 98.3 ± 0.64 | 81.7 ± 11.7 | 92.0 ± 2.00 | 66.9 ± 27.7 | 96.1 ± 2.68 | 14.6 ± 3.61 | 74.9 ± 4.83 |
| | ERNIE | 98.6 ± 0.58 | 85.7 ± 5.51 | 75.0 ± 14.8 | 85.4 ± 12.6 | 93.8 ± 3.13 | 7.30 ± 3.61 | 74.6 ± 3.66 |
| | ROMANCE | 98.4 ± 1.69 | 88.3 ± 1.73 | 84.7 ± 4.62 | 92.0 ± 5.57 | 97.2 ± 1.37 | 39.2 ± 4.02 | 83.3 ± 1.69 |
| | WALL | 98.8 ± 1.00 | **90.0 ± 3.61** | 90.7 ± 1.53 | 96.2 ± 2.20 | 98.8 ± 0.17 | 26.2 ± 18.1 | 83.5 ± 3.08 |
| | IBAL (ours) | **99.5 ± 0.53** | 89.0 ± 2.52 | 92.3 ± 1.53 | **97.8 ± 1.08** | **98.9 ± 1.74** | 91.7 ± 3.61 | **94.8 ± 0.85** |
| 2s3z | Vanilla QMIX | 97.6 ± 0.65 | 75.0 ± 7.21 | 58.3 ± 7.37 | 53.8 ± 4.51 | 39.2 ± 6.88 | 3.10 ± 0.00 | 54.7 ± 2.41 |
| | Rand-Obs | 98.2 ± 1.48 | 80.0 ± 1.00 | 89.3 ± 2.52 | 53.1 ± 10.8 | 52.1 ± 10.0 | 5.20 ± 6.50 | 63.0 ± 3.06 |
| | Rand-Act | 96.1 ± 2.65 | 79.0 ± 6.93 | 77.3 ± 3.21 | 68.7 ± 5.47 | 62.6 ± 14.2 | 4.10 ± 1.79 | 64.7 ± 2.97 |
| | FGSM | 96.5 ± 2.55 | 85.7 ± 3.51 | **95.7 ± 2.31** | 62.5 ± 14.3 | 61.4 ± 13.0 | 6.20 ± 5.41 | 68.0 ± 3.77 |
| | ATLA | 96.6 ± 5.21 | 80.3 ± 2.08 | 89.7 ± 4.16 | 54.2 ± 18.0 | 53.1 ± 5.42 | 9.40 ± 8.27 | 63.9 ± 4.65 |
| | ERNIE | 97.6 ± 3.35 | 82.3 ± 6.66 | 74.7 ± 9.29 | 63.5 ± 4.77 | 49.2 ± 6.88 | 4.20 ± 1.80 | 61.9 ± 2.94 |
| | ROMANCE | 97.9 ± 1.69 | 82.0 ± 4.36 | 76.7 ± 1.53 | 83.3 ± 10.0 | 65.0 ± 3.48 | 13.3 ± 7.22 | 69.7 ± 2.52 |
| | WALL | 98.9 ± 0.35 | 91.7 ± 0.58 | 86.7 ± 8.51 | 87.5 ± 0.98 | **92.7 ± 4.78** | 13.5 ± 1.80 | 78.9 ± 1.93 |
| | IBAL (ours) | **99.5 ± 1.75** | **93.3 ± 0.58** | 89.8 ± 1.00 | 89.8 ± 3.44 | 83.4 ± 4.91 | 78.7 ± 3.51 | **89.1 ± 1.21** |

*Table 8.* Success rates (%) under various adversarial attacks on SMAC scenarios: `8m`, `1c3s5z`, and `MMM`.

| Scenario | Model | Attack | | | | | | Mean |
|---|---|---|---|---|---|---|---|---|
| | | Nat. | Rand. | FGSM | EGA | Wolf. | Ours | |
| 8m | Vanilla QMIX | 92.8 ± 1.88 | 83.0 ± 2.45 | 75.7 ± 3.47 | 70.8 ± 16.0 | 14.6 ± 4.48 | 7.30 ± 3.61 | 57.3 ± 3.08 |
| | Rand-Obs | 98.0 ± 1.06 | 85.6 ± 5.41 | 88.8 ± 3.12 | 74.0 ± 13.0 | 29.2 ± 3.61 | 6.20 ± 6.25 | 63.6 ± 3.39 |
| | Rand-Act | 97.6 ± 1.19 | 84.0 ± 1.57 | 79.1 ± 3.83 | 85.2 ± 6.41 | 28.1 ± 10.8 | 6.20 ± 8.25 | 63.4 ± 2.90 |
| | FGSM | 98.0 ± 1.06 | 90.7 ± 3.13 | **96.5 ± 2.64** | 86.4 ± 5.09 | 41.7 ± 4.78 | 10.4 ± 1.80 | 70.3 ± 1.68 |
| | ATLA | 92.7 ± 3.61 | 80.0 ± 13.0 | 90.3 ± 2.31 | 74.0 ± 3.61 | 39.6 ± 4.77 | 15.6 ± 12.5 | 65.4 ± 3.61 |
| | ERNIE | 96.9 ± 3.12 | 84.4 ± 3.13 | 90.7 ± 5.77 | 83.4 ± 6.66 | 26.7 ± 4.43 | 6.20 ± 5.41 | 64.2 ± 2.78 |
| | ROMANCE | 98.1 ± 1.00 | 89.8 ± 3.45 | 90.9 ± 4.48 | 90.0 ± 3.81 | 33.3 ± 5.73 | 11.0 ± 11.0 | 68.8 ± 2.79 |
| | WALL | 96.5 ± 5.12 | 92.2 ± 1.86 | 92.7 ± 4.77 | 93.7 ± 3.21 | **85.5 ± 1.95** | 13.5 ± 7.86 | 79.0 ± 2.15 |
| | IBAL (ours) | **99.3 ± 1.15** | **94.5 ± 3.38** | 95.6 ± 4.68 | **97.6 ± 1.22** | 79.0 ± 4.00 | **88.4 ± 3.32** | **92.4 ± 1.33** |
| 1c3s5z | Vanilla QMIX | 98.8 ± 0.75 | 84.7 ± 2.08 | 81.7 ± 2.89 | 83.8 ± 0.97 | 52.7 ± 7.22 | 18.8 ± 8.27 | 70.7 ± 1.95 |
| | Rand-Obs | 96.8 ± 3.03 | 89.0 ± 1.00 | 86.3 ± 0.58 | 86.8 ± 1.16 | 56.1 ± 9.33 | 7.30 ± 6.50 | 70.9 ± 1.88 |
| | Rand-Act | 98.3 ± 1.23 | 90.0 ± 5.20 | 83.3 ± 0.58 | 86.5 ± 3.61 | 64.8 ± 3.63 | 6.30 ± 5.42 | 71.5 ± 1.34 |
| | FGSM | 98.6 ± 1.51 | 92.7 ± 2.52 | 97.0 ± 1.00 | 88.5 ± 2.91 | 67.6 ± 7.54 | 6.30 ± 3.12 | 75.8 ± 1.45 |
| | ATLA | 96.9 ± 0.01 | 94.0 ± 0.00 | 94.7 ± 0.58 | 84.4 ± 3.13 | 73.1 ± 4.10 | 24.0 ± 10.0 | 77.1 ± 0.86 |
| | ERNIE | 94.8 ± 6.37 | 93.3 ± 4.04 | 93.0 ± 0.00 | 88.5 ± 1.81 | 80.6 ± 11.9 | 17.7 ± 3.62 | 77.2 ± 2.46 |
| | ROMANCE | **99.1 ± 0.65** | 94.3 ± 0.58 | 91.0 ± 2.65 | 89.6 ± 4.77 | 89.2 ± 2.01 | 12.5 ± 5.41 | 79.2 ± 1.12 |
| | WALL | 98.9 ± 0.95 | 96.3 ± 3.06 | 96.7 ± 1.53 | 93.8 ± 3.13 | 95.8 ± 1.81 | 18.8 ± 6.25 | 83.4 ± 1.51 |
| | IBAL (ours) | 99.0 ± 1.80 | **97.3 ± 2.52** | **98.3 ± 2.89** | **93.9 ± 3.03** | **98.9 ± 1.95** | **91.3 ± 3.80** | **96.4 ± 1.12** |
| MMM | Vanilla QMIX | 98.6 ± 1.51 | 81.3 ± 9.07 | 83.3 ± 5.51 | 65.6 ± 16.5 | 29.5 ± 21.7 | 0.00 ± 0.00 | 59.0 ± 5.59 |
| | Rand-Obs | 96.5 ± 2.64 | 93.0 ± 2.00 | 93.7 ± 0.46 | 80.2 ± 4.78 | 59.9 ± 8.33 | 0.00 ± 0.00 | 70.6 ± 1.71 |
| | Rand-Act | 97.5 ± 1.11 | 93.7 ± 1.53 | 90.7 ± 1.53 | 85.4 ± 3.61 | 66.6 ± 12.6 | 0.00 ± 0.00 | 72.3 ± 2.14 |
| | FGSM | 98.0 ± 1.06 | 92.3 ± 1.53 | **98.0 ± 2.00** | 73.8 ± 12.8 | 57.9 ± 9.48 | 7.30 ± 10.0 | 71.2 ± 2.94 |
| | ATLA | 97.5 ± 3.29 | 94.3 ± 1.53 | 95.7 ± 3.51 | 85.4 ± 6.50 | 72.3 ± 4.19 | 3.10 ± 5.41 | 74.7 ± 1.74 |
| | ERNIE | 98.0 ± 1.06 | 94.7 ± 1.53 | 94.3 ± 2.08 | 69.8 ± 14.8 | 41.2 ± 12.9 | 2.10 ± 3.61 | 66.3 ± 3.93 |
| | ROMANCE | 94.8 ± 3.61 | 92.7 ± 6.81 | 90.3 ± 10.0 | 88.5 ± 11.8 | 57.2 ± 15.5 | 4.20 ± 1.80 | 62.9 ± 3.63 |
| | WALL | 95.9 ± 4.57 | 93.0 ± 10.4 | 96.0 ± 5.29 | 92.4 ± 9.69 | **95.6 ± 2.96** | 13.5 ± 18.3 | 81.1 ± 4.62 |
| | IBAL (ours) | **99.7 ± 0.58** | **98.3 ± 0.58** | 97.0 ± 1.00 | **92.7 ± 5.01** | 84.4 ± 3.97 | **88.7 ± 7.23** | **93.5 ± 1.62** |

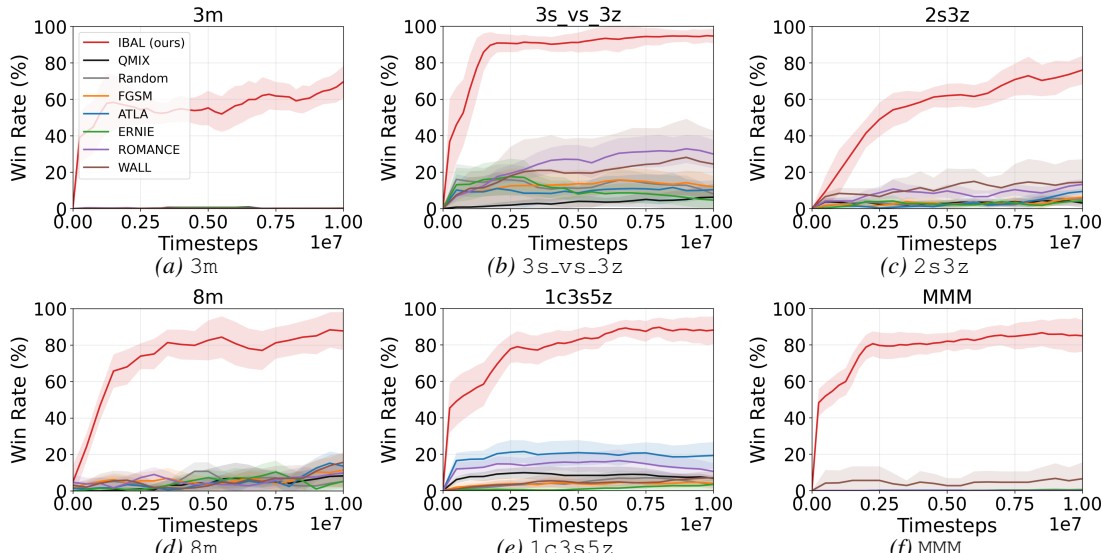

*Figure 13.* Performance comparison under *unseen* interaction-breaking attack.

## E.2. Full Performance Comparison Result under Non-Parametric Perturbations

Table 9, 10 report the full results with standard deviations under non-parametric perturbations, including Dis-$\ell$ with $\ell \in \{1, 2\}$ and HP-$h$ with $h \in \{10, 15, 20\}$. For the 3m, 3s_vs_3z, 2s3z, and 8m scenarios, we report results under Dis-1 only, since no method can solve the Dis-2 setting on these maps. For the remaining scenarios, we report both Dis-1 and Dis-2. Across these settings, IBAL consistently outperforms all baselines. In Dis-$\ell$, most methods degrade markedly when ally units are disabled, whereas IBAL remains effective, suggesting more resilient coordination under partial interaction failures. A similar trend holds for HP-$h$, where IBAL achieves substantially higher win rates across all thresholds. Overall, these results indicate that training under diverse interaction-breaking conditions improves robustness to non-parametric perturbations.

*Table 9.* Success rates (%) under non-parametric perturbations on SMAC scenarios: 3m, 3s_vs_3z, 2s3z, and 8m.

| Scenario | Model | Attack | | | | Mean |
|---|---|---|---|---|---|---|
| | | Dis-1 | HP-10 | HP-15 | HP-20 | |
| 3m | Vanilla QMIX | $0.00 \pm 0.00$ | $56.3 \pm 6.43$ | $62.0 \pm 2.00$ | $1.70 \pm 2.08$ | $30.5 \pm 2.13$ |
| | Rand-Obs | $0.00 \pm 0.00$ | $62.0 \pm 10.4$ | $70.3 \pm 9.75$ | $\mathbf{6.00 \pm 9.54}$ | $34.1 \pm 7.42$ |
| | Rand-Act | $0.00 \pm 0.00$ | $41.7 \pm 11.0$ | $66.7 \pm 9.02$ | $0.00 \pm 0.00$ | $27.1 \pm 7.74$ |
| | FGSM | $1.00 \pm 1.80$ | $70.7 \pm 6.81$ | $\underline{81.2 \pm 6.25}$ | $4.30 \pm 2.52$ | $\underline{39.3 \pm 4.34}$ |
| | ATLA | $0.00 \pm 0.00$ | $\mathbf{77.3 \pm 0.58}$ | $78.1 \pm 10.8$ | $\underline{4.70 \pm 0.58}$ | $39.0 \pm 2.95$ |
| | ERNIE | $\underline{3.10 \pm 5.41}$ | $62.7 \pm 6.43$ | $74.0 \pm 11.0$ | $1.30 \pm 0.58$ | $35.9 \pm 5.70$ |
| | ROMANCE | $0.00 \pm 0.00$ | $64.7 \pm 9.07$ | $70.8 \pm 23.7$ | $3.30 \pm 3.06$ | $34.7 \pm 8.94$ |
| | WALL | $0.00 \pm 0.00$ | $70.0 \pm 16.1$ | $67.7 \pm 7.86$ | $4.00 \pm 2.65$ | $35.4 \pm 6.65$ |
| | IBAL (ours) | $\mathbf{25.0 \pm 3.71}$ | $76.0 \pm 5.20$ | $\mathbf{83.8 \pm 5.22}$ | $1.70 \pm 2.07$ | $\mathbf{46.6 \pm 4.58}$ |
| 3s_vs_3z | Vanilla QMIX | $5.20 \pm 6.50$ | $89.0 \pm 1.00$ | $76.7 \pm 11.59$ | $2.10 \pm 3.61$ | $43.2 \pm 5.68$ |
| | Rand-Obs | $22.1 \pm 2.75$ | $96.3 \pm 2.31$ | $88.1 \pm 2.14$ | $29.2 \pm 7.22$ | $58.9 \pm 3.60$ |
| | Rand-Act | $12.5 \pm 5.41$ | $90.7 \pm 8.08$ | $91.3 \pm 2.17$ | $30.2 \pm 3.62$ | $56.2 \pm 4.82$ |
| | FGSM | $25.0 \pm 16.5$ | $95.8 \pm 5.03$ | $96.2 \pm 1.57$ | $41.2 \pm 10.3$ | $64.6 \pm 8.37$ |
| | ATLA | $3.10 \pm 3.12$ | $98.2 \pm 0.76$ | $96.4 \pm 2.32$ | $43.8 \pm 5.41$ | $60.9 \pm 2.90$ |
| | ERNIE | $6.20 \pm 10.8$ | $98.0 \pm 0.97$ | $92.1 \pm 4.02$ | $45.4 \pm 25.9$ | $60.4 \pm 10.4$ |
| | ROMANCE | $\underline{55.4 \pm 3.55}$ | $98.0 \pm 2.65$ | $96.8 \pm 1.75$ | $57.1 \pm 27.9$ | $\underline{76.1 \pm 8.96}$ |
| | WALL | $37.5 \pm 19.0$ | $\underline{98.8 \pm 0.75}$ | $\underline{97.8 \pm 1.59}$ | $\underline{68.8 \pm 11.3}$ | $75.7 \pm 8.16$ |
| | IBAL (ours) | $\mathbf{93.5 \pm 2.94}$ | $\mathbf{99.7 \pm 0.58}$ | $\mathbf{99.9 \pm 0.10}$ | $\mathbf{81.7 \pm 6.13}$ | $\mathbf{93.7 \pm 2.44}$ |
| 2s3z | Vanilla QMIX | $0.00 \pm 0.00$ | $80.7 \pm 10.1$ | $46.2 \pm 19.8$ | $33.0 \pm 6.08$ | $39.9 \pm 9.95$ |
| | Rand-Obs | $3.10 \pm 3.12$ | $84.3 \pm 6.51$ | $58.3 \pm 17.2$ | $28.0 \pm 5.00$ | $43.4 \pm 7.21$ |
| | Rand-Act | $2.10 \pm 1.79$ | $81.0 \pm 5.20$ | $46.9 \pm 5.40$ | $33.3 \pm 11.0$ | $40.8 \pm 5.78$ |
| | FGSM | $1.00 \pm 1.80$ | $84.0 \pm 2.00$ | $62.5 \pm 8.27$ | $32.0 \pm 1.73$ | $44.9 \pm 3.45$ |
| | ATLA | $4.20 \pm 7.22$ | $82.0 \pm 1.73$ | $54.2 \pm 14.4$ | $28.0 \pm 9.64$ | $42.1 \pm 8.25$ |
| | ERNIE | $0.00 \pm 0.00$ | $86.3 \pm 0.58$ | $63.5 \pm 3.61$ | $28.3 \pm 4.73$ | $44.5 \pm 1.98$ |
| | ROMANCE | $\underline{8.30 \pm 3.61}$ | $83.7 \pm 2.52$ | $60.4 \pm 17.2$ | $41.7 \pm 9.45$ | $48.5 \pm 8.95$ |
| | WALL | $7.30 \pm 6.50$ | $\underline{88.3 \pm 2.31}$ | $\underline{75.0 \pm 8.27}$ | $\mathbf{60.7 \pm 5.03}$ | $\underline{57.8 \pm 5.52}$ |
| | IBAL (ours) | $\mathbf{81.7 \pm 4.51}$ | $\mathbf{92.7 \pm 6.03}$ | $\mathbf{78.0 \pm 4.36}$ | $\underline{59.3 \pm 16.3}$ | $\mathbf{77.9 \pm 7.79}$ |
| 8m | Vanilla QMIX | $13.3 \pm 1.44$ | $54.3 \pm 6.03$ | $52.5 \pm 12.1$ | $0.00 \pm 0.00$ | $30.0 \pm 4.89$ |
| | Rand-Obs | $10.4 \pm 1.75$ | $42.3 \pm 3.06$ | $43.8 \pm 3.12$ | $0.00 \pm 0.00$ | $24.1 \pm 1.98$ |
| | Rand-Act | $22.3 \pm 6.88$ | $34.0 \pm 13.1$ | $41.2 \pm 20.1$ | $0.00 \pm 0.00$ | $24.4 \pm 10.1$ |
| | FGSM | $21.9 \pm 6.25$ | $58.0 \pm 17.1$ | $64.6 \pm 20.1$ | $1.00 \pm 1.73$ | $36.4 \pm 11.3$ |
| | ATLA | $5.20 \pm 1.80$ | $45.7 \pm 13.3$ | $50.0 \pm 16.2$ | $0.00 \pm 0.00$ | $25.2 \pm 7.83$ |
| | ERNIE | $8.30 \pm 7.22$ | $32.7 \pm 17.2$ | $53.1 \pm 9.38$ | $0.00 \pm 0.00$ | $23.5 \pm 8.46$ |
| | ROMANCE | $18.8 \pm 15.6$ | $62.0 \pm 12.1$ | $58.3 \pm 15.7$ | $0.70 \pm 0.58$ | $35.9 \pm 11.0$ |
| | WALL | $\underline{25.4 \pm 13.1}$ | $\underline{66.7 \pm 14.3}$ | $\underline{70.8 \pm 10.0}$ | $\underline{1.70 \pm 2.08}$ | $\underline{41.1 \pm 9.38}$ |
| | IBAL (ours) | $\mathbf{87.0 \pm 5.17}$ | $\mathbf{92.3 \pm 3.79}$ | $\mathbf{89.8 \pm 7.36}$ | $\mathbf{20.0 \pm 4.36}$ | $\mathbf{72.3 \pm 5.17}$ |

*Table 10.* Success rates (%) under non-parametric perturbations on SMAC scenarios: `1c3s5z` and `MMM`.

| Scenario | Model | Attack | | | | | Mean |
|---|---|---|---|---|---|---|---|
| | | Dis-1 | Dis-2 | HP-10 | HP-15 | HP-20 | |
| `1c3s5z` | Vanilla QMIX | 80.8 ± 3.73 | 35.4 ± 10.9 | 85.3 ± 4.51 | 84.3 ± 4.51 | 66.3 ± 4.62 | 70.4 ± 5.67 |
| | Rand-Obs | 81.2 ± 8.27 | 42.0 ± 17.5 | 83.7 ± 1.16 | 77.0 ± 7.81 | 67.0 ± 10.4 | 70.2 ± 9.04 |
| | Rand-Act | 80.6 ± 10.5 | 26.3 ± 6.62 | 85.7 ± 2.89 | 70.0 ± 8.66 | 61.5 ± 8.19 | 64.8 ± 7.38 |
| | FGSM | 80.2 ± 11.8 | 32.3 ± 14.0 | 88.0 ± 5.00 | 82.7 ± 9.24 | 67.9 ± 11.2 | 70.2 ± 10.2 |
| | ATLA | 85.4 ± 4.77 | 39.6 ± 24.3 | 90.3 ± 1.16 | 78.3 ± 6.51 | 62.0 ± 8.00 | 71.1 ± 8.94 |
| | ERNIE | 87.6 ± 3.01 | 29.5 ± 6.95 | 91.0 ± 6.93 | 79.3 ± 14.4 | 47.7 ± 45.2 | 67.0 ± 15.3 |
| | ROMANCE | 86.5 ± 7.22 | 33.3 ± 7.22 | 88.0 ± 3.61 | 72.7 ± 13.8 | 58.7 ± 5.77 | 67.8 ± 7.84 |
| | WALL | 80.2 ± 6.50 | 50.0 ± 13.6 | 92.0 ± 2.65 | 81.7 ± 10.8 | 61.7 ± 16.8 | 73.1 ± 12.1 |
| | IBAL (ours) | **99.0 ± 1.80** | **71.0 ± 4.77** | **93.3 ± 3.22** | **86.7 ± 7.51** | **83.3 ± 4.93** | **86.7 ± 4.45** |
| `MMM` | Vanilla QMIX | 51.0 ± 4.77 | 1.00 ± 1.80 | 87.7 ± 3.79 | 74.0 ± 8.54 | 39.6 ± 16.0 | 50.2 ± 6.99 |
| | Rand-Obs | 71.9 ± 6.25 | 8.30 ± 4.77 | 93.0 ± 3.61 | 88.3 ± 5.50 | 65.4 ± 11.2 | 65.4 ± 5.65 |
| | Rand-Act | 81.2 ± 8.27 | 10.4 ± 3.61 | 91.7 ± 5.13 | 84.0 ± 6.08 | 58.3 ± 7.22 | 65.1 ± 5.47 |
| | FGSM | 67.3 ± 7.94 | 3.10 ± 3.12 | **95.0 ± 1.73** | 77.0 ± 9.85 | 52.1 ± 14.1 | 58.9 ± 7.35 |
| | ATLA | 65.6 ± 11.3 | 8.30 ± 9.02 | 91.7 ± 6.66 | 83.7 ± 8.96 | 51.0 ± 15.4 | 60.1 ± 10.3 |
| | ERNIE | 69.8 ± 14.4 | 7.30 ± 3.61 | 89.3 ± 7.37 | 77.0 ± 16.5 | 49.0 ± 13.0 | 58.5 ± 11.0 |
| | ROMANCE | 71.9 ± 9.38 | 13.5 ± 7.86 | 91.0 ± 8.18 | 79.3 ± 3.21 | 61.5 ± 11.8 | 63.4 ± 8.89 |
| | WALL | 77.1 ± 19.1 | 27.1 ± 4.77 | 90.0 ± 16.5 | 83.7 ± 21.5 | 82.3 ± 12.6 | 72.1 ± 14.9 |
| | IBAL (ours) | **92.7 ± 2.31** | **58.1 ± 8.72** | 94.3 ± 2.52 | **89.7 ± 3.21** | **86.7 ± 1.44** | **84.3 ± 3.24** |

### E.3. Performance Comparison Results on Additional Benchmarks

**Level-Based Foraging.** Table 11 reports additional success-rate comparisons on Level-Based Foraging (LBF) using the `Foraging-20x20-np-6f-s3` scenario, where $n$ denotes the number of agents and we evaluate 5, 10, and 15-agent settings. Existing attack methods have only limited effects on LBF, as most baselines still maintain relatively high success rates under random and FGSM attacks. In contrast, the proposed interaction-breaking attack is substantially more damaging because it directly disrupts inter-agent coordination. IBAL remains robust not only under the proposed attack but also across other attack settings, consistently achieving strong performance across different numbers of agents. These results suggest that interaction-breaking adversarial learning transfers effectively to cooperative foraging tasks beyond SMAC.

*Table 11.* Success rates (%) on Level-Based Foraging (LBF) under various attack settings.

| Scenario | Model | Attack | | | |
|---|---|---|---|---|---|
| | | Nat. | Rand. | FGSM | Ours |
| 5p | Vanilla QMIX | 80.0 ± 1.9 | 74.0 ± 1.4 | 69.0 ± 6.4 | 48.0 ± 4.9 |
| | Rand-Obs | 80.0 ± 9.8 | 77.0 ± 0.7 | 72.0 ± 7.1 | 47.0 ± 1.4 |
| | Rand-Act | 80.0 ± 5.8 | 77.0 ± 3.3 | 72.0 ± 1.8 | 46.0 ± 5.7 |
| | FGSM | 77.0 ± 6.9 | 73.0 ± 3.7 | 77.0 ± 3.0 | 57.0 ± 3.5 |
| | ATLA | 73.0 ± 3.4 | 72.0 ± 2.6 | 72.0 ± 7.1 | 55.0 ± 1.4 |
| | IBAL (ours) | **83.0 ± 1.9** | **81.0 ± 0.9** | **78.0 ± 2.0** | **70.0 ± 2.1** |
| 10p | Vanilla QMIX | **97.0 ± 2.1** | 92.0 ± 3.0 | 89.0 ± 4.9 | 56.0 ± 2.3 |
| | Rand-Obs | 93.0 ± 1.1 | 94.0 ± 1.6 | 90.0 ± 2.9 | 55.0 ± 0.1 |
| | Rand-Act | 95.0 ± 0.6 | 92.0 ± 0.6 | 90.0 ± 3.7 | 63.0 ± 4.2 |
| | FGSM | 92.0 ± 1.8 | 94.0 ± 4.2 | 94.0 ± 0.3 | 76.0 ± 1.0 |
| | ATLA | 94.0 ± 0.7 | 94.0 ± 1.4 | 93.0 ± 0.1 | 50.0 ± 1.7 |
| | IBAL (ours) | 96.0 ± 1.4 | **97.0 ± 0.5** | **95.0 ± 0.5** | **95.0 ± 0.4** |
| 15p | Vanilla QMIX | 96.0 ± 1.9 | 93.0 ± 1.5 | 92.0 ± 0.5 | 54.0 ± 3.9 |
| | Rand-Obs | 96.0 ± 1.3 | 96.0 ± 0.1 | 96.0 ± 1.3 | 67.0 ± 8.2 |
| | Rand-Act | **99.0 ± 0.1** | 97.0 ± 0.1 | 96.0 ± 1.4 | 67.0 ± 9.2 |
| | FGSM | 99.0 ± 0.2 | 97.0 ± 0.7 | **98.0 ± 2.0** | 65.0 ± 1.5 |
| | ATLA | 98.0 ± 0.8 | 94.0 ± 0.7 | 96.0 ± 0.3 | 66.0 ± 0.6 |
| | IBAL (ours) | 99.0 ± 1.1 | **99.0 ± 1.4** | 98.0 ± 2.1 | **96.0 ± 0.2** |

**SMACv2.** Table 12 reports additional win-rate comparisons on three SMACv2 scenarios, `terran_5_vs_5`, `zerg_5_vs_5`, and `protoss_5_vs_5`, which introduce greater stochasticity than SMAC. Due to this increased stochasticity, all methods generally achieve lower natural performance than in the original SMAC scenarios, indicating that SMACv2 poses a more challenging coordination problem even without explicit attacks. Nevertheless, IBAL achieves the highest natural performance across all three SMACv2 scenarios, suggesting that interaction-breaking adversarial learning improves robustness not only against explicit attacks but also against stochastic variations in the environment. Under the proposed interaction-breaking attack, baseline methods suffer severe performance drops, whereas IBAL consistently preserves substantially higher win rates. These results indicate that IBAL learns policies that are less dependent on brittle inter-agent interactions and therefore remain more reliable both under attacks and in highly stochastic cooperative combat scenarios.

*Table 12.* Win rates (%) on SMACv2 scenarios under various attack settings.

| Scenario | Model | Attack | | | |
|---|---|---|---|---|---|
| | | Nat. | Rand. | FGSM | Ours |
| `terran_5_vs_5` | Vanilla QMIX | $60.9 \pm 2.20$ | $55.6 \pm 0.900$ | $39.1 \pm 2.20$ | $10.9 \pm 6.60$ |
| | Rand-Obs | $64.1 \pm 6.60$ | $66.2 \pm 3.70$ | $48.4 \pm 2.20$ | $12.5 \pm 4.40$ |
| | Rand-Act | $71.9 \pm 4.40$ | $62.3 \pm 5.00$ | $50.0 \pm 8.80$ | $26.6 \pm 2.20$ |
| | FGSM | $60.9 \pm 2.20$ | $60.9 \pm 15.5$ | $57.8 \pm 2.20$ | $17.2 \pm 11.0$ |
| | ATLA | $65.9 \pm 8.40$ | $57.8 \pm 11.0$ | $56.3 \pm 8.80$ | $20.9 \pm 1.30$ |
| | IBAL (ours) | $\mathbf{78.1 \pm 4.40}$ | $\mathbf{72.2 \pm 4.90}$ | $\mathbf{64.7 \pm 3.10}$ | $\mathbf{48.8 \pm 2.50}$ |
| `zerg_5_vs_5` | Vanilla QMIX | $45.3 \pm 11.0$ | $24.7 \pm 4.90$ | $21.9 \pm 4.40$ | $11.3 \pm 1.80$ |
| | Rand-Obs | $48.4 \pm 2.20$ | $45.3 \pm 11.0$ | $24.9 \pm 4.40$ | $15.4 \pm 9.10$ |
| | Rand-Act | $35.9 \pm 2.20$ | $43.8 \pm 8.80$ | $29.7 \pm 2.20$ | $18.7 \pm 4.40$ |
| | FGSM | $40.6 \pm 4.50$ | $42.8 \pm 1.40$ | $40.6 \pm 17.8$ | $26.5 \pm 2.10$ |
| | ATLA | $42.2 \pm 2.20$ | $36.5 \pm 1.40$ | $26.6 \pm 6.60$ | $16.5 \pm 2.10$ |
| | IBAL (ours) | $\mathbf{51.6 \pm 11.1}$ | $\mathbf{46.9 \pm 4.40}$ | $\mathbf{42.2 \pm 15.5}$ | $\mathbf{33.1 \pm 2.80}$ |
| `protoss_5_vs_5` | Vanilla QMIX | $54.7 \pm 20.8$ | $37.5 \pm 8.90$ | $39.1 \pm 15.5$ | $5.4 \pm 1.20$ |
| | Rand-Obs | $62.2 \pm 4.90$ | $70.3 \pm 11.0$ | $35.9 \pm 11.1$ | $21.9 \pm 4.40$ |
| | Rand-Act | $59.4 \pm 8.80$ | $65.6 \pm 4.40$ | $45.3 \pm 15.5$ | $18.4 \pm 0.400$ |
| | FGSM | $56.9 \pm 17.7$ | $48.4 \pm 6.60$ | $\mathbf{70.3 \pm 2.20}$ | $7.8 \pm 2.20$ |
| | ATLA | $59.4 \pm 13.3$ | $51.6 \pm 2.20$ | $43.8 \pm 2.60$ | $4.6 \pm 2.30$ |
| | IBAL (ours) | $\mathbf{77.2 \pm 2.20}$ | $\mathbf{75.0 \pm 4.40}$ | $61.9 \pm 8.00$ | $\mathbf{50.4 \pm 2.00}$ |

### E.4. Performance Comparison Results on Other CTDE Algorithms

Table 13 reports additional performance comparisons using MAPPO as another CTDE backbone. Although the main experiments use QMIX, IBAL does not rely on value information or the IGM property for constructing attacks, unlike many existing attack-based robust MARL methods. Therefore, IBAL is conceptually applicable to a broader class of MARL algorithms beyond value-decomposition methods. The results show that MAPPO+IBAL maintains strong performance and robustness across all tested scenarios, especially under the proposed interaction-breaking attack, where other MAPPO-based baselines suffer severe performance degradation. This supports that the effectiveness of IBAL is not tied to a specific MARL algorithm and can also improve robustness for policy-gradient-based CTDE methods such as MAPPO.

*Table 13.* Win rates (%) with MAPPO-based CTDE algorithms under various attack settings.

| Scenario | Model | Attack | | |
|---|---|---|---|---|
| | | Nat. | Rand. | Ours |
| 3s_vs_3z | MAPPO | $98.0 \pm 2.8$ | $75.4 \pm 4.4$ | $4.7 \pm 2.2$ |
| | MAPPO+Rand-Obs | $\underline{99.0 \pm 1.4}$ | $74.5 \pm 5.0$ | $1.6 \pm 2.2$ |
| | MAPPO+Rand-Act | $97.4 \pm 0.8$ | $82.8 \pm 6.6$ | $4.7 \pm 2.2$ |
| | MAPPO+FGSM | $\mathbf{99.0 \pm 1.4}$ | $\underline{95.0 \pm 7.1}$ | $\underline{17.2 \pm 11.0}$ |
| | MAPPO+ATLA | $95.5 \pm 2.4$ | $82.8 \pm 6.6$ | $0.0 \pm 0.0$ |
| | MAPPO+IBAL | $98.5 \pm 2.1$ | $\mathbf{99.0 \pm 1.4}$ | $\mathbf{97.4 \pm 0.8}$ |
| 2s3z | MAPPO | $97.9 \pm 2.7$ | $81.8 \pm 8.1$ | $11.3 \pm 1.4$ |
| | MAPPO+Rand-Obs | $98.4 \pm 2.2$ | $84.4 \pm 4.4$ | $16.8 \pm 6.0$ |
| | MAPPO+Rand-Act | $98.0 \pm 2.8$ | $89.1 \pm 2.2$ | $20.9 \pm 1.3$ |
| | MAPPO+FGSM | $\underline{99.5 \pm 0.6}$ | $\underline{95.9 \pm 1.2}$ | $\underline{45.3 \pm 2.2}$ |
| | MAPPO+ATLA | $99.5 \pm 0.6$ | $95.3 \pm 2.2$ | $26.6 \pm 11.0$ |
| | MAPPO+IBAL | $\mathbf{100.0 \pm 0.0}$ | $\mathbf{96.9 \pm 0.1}$ | $\mathbf{95.3 \pm 2.2}$ |
| 8m | MAPPO | $98.5 \pm 0.7$ | $76.6 \pm 2.2$ | $4.7 \pm 6.6$ |
| | MAPPO+Rand-Obs | $97.4 \pm 0.8$ | $97.5 \pm 3.5$ | $\underline{25.0 \pm 4.4}$ |
| | MAPPO+Rand-Act | $95.9 \pm 1.3$ | $\underline{98.1 \pm 0.1}$ | $18.6 \pm 4.6$ |
| | MAPPO+FGSM | $\underline{99.0 \pm 1.4}$ | $95.3 \pm 2.2$ | $23.4 \pm 6.6$ |
| | MAPPO+ATLA | $\mathbf{100.0 \pm 0.0}$ | $92.2 \pm 2.2$ | $18.8 \pm 17.7$ |
| | MAPPO+IBAL | $98.3 \pm 2.5$ | $\mathbf{98.4 \pm 2.2}$ | $\mathbf{95.0 \pm 2.8}$ |

# F. Additional Analyses for IBAL

This section provides additional analyses and ablation studies of IBAL. Appendix F.1 analyzes the group redundancy term introduced in Lemma 4.3 on SMAC environments. Appendix F.2 examines IBAL's behavior under the disabled setting. Appendix F.3 presents further ablation studies on the masking budget $L$ and the minimum attack probability. Finally, Appendix F.4 discusses computational complexity.

## F.1. Analysis of Group Redundancy

Lemma 4.3 uses a *group redundancy term* to decompose a observation-level MI term. We empirically find that this redundancy is negligible in SMAC. This indicates that the dimension-wise MI closely approximates the corresponding group-wise MI in practice, thereby supporting the validity of MI-based dimension selection. We focus on 3m and 3s_vs_3z, where the number of agents is small and MI estimation is comparatively tractable.

Following Appendix A.2 and (7), the group redundancy term is defined as

$$\mathcal{R}(G_1; G_2) = \underbrace{\mathcal{I}\Big(\tilde{\boldsymbol{o}}_{t+1}^{G_1}; \boldsymbol{a}_t^{G_2} \,\big|\, \boldsymbol{a}_t^{G_1}, \boldsymbol{\tau}_t\Big)}_{\textbf{Group-wise MI}} - \underbrace{\sum_{i \in G_1} \sum_{j \in G_2} \sum_{d=1}^{d_o} \mathcal{I}\Big(\tilde{o}_{d,t+1}^{i}; \boldsymbol{a}_t^{j} \,\big|\, \boldsymbol{a}_t^{i}, \boldsymbol{\tau}_t\Big)}_{\textbf{Individual MI}} .$$

Fig. 14 plots the per-timestep values of the group redundancy, the group-wise MI, and the individual MI. We observe that (a) the group redundancy stays close to zero and its magnitude is substantially smaller than the MI values in (b) and (c), suggesting that it has negligible impact on dimension-wise MI selection. Overall, these results empirically support the validity of decomposing the observation-level MI term into dimension-wise components in practice.

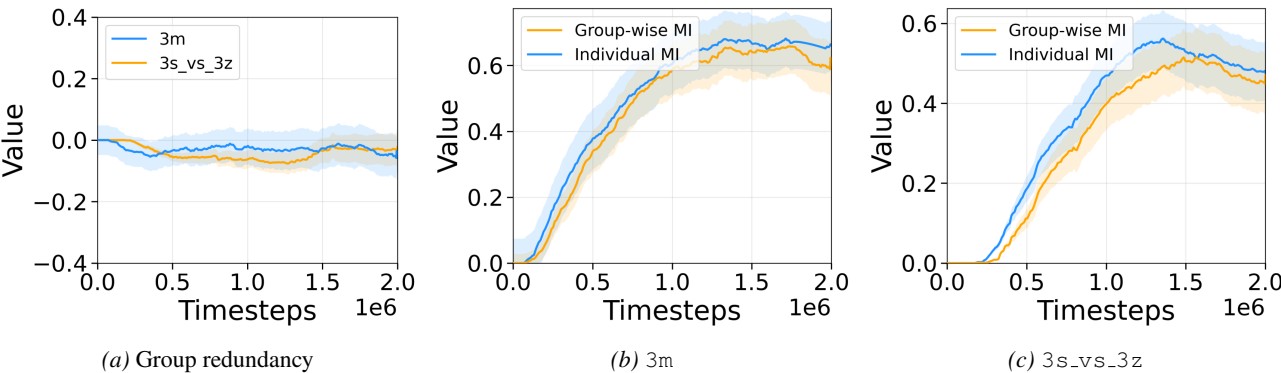

*(a)* Group redundancy    *(b)* 3m    *(c)* 3s_vs_3z

*Figure 14.* Comparison results: (a) group redundancy on 3m and 3s_vs_3z, and group-wise/individual MI on (b) 3m and (c) 3s_vs_3z

### F.2. Trajectory Analysis under the Disabled Setting

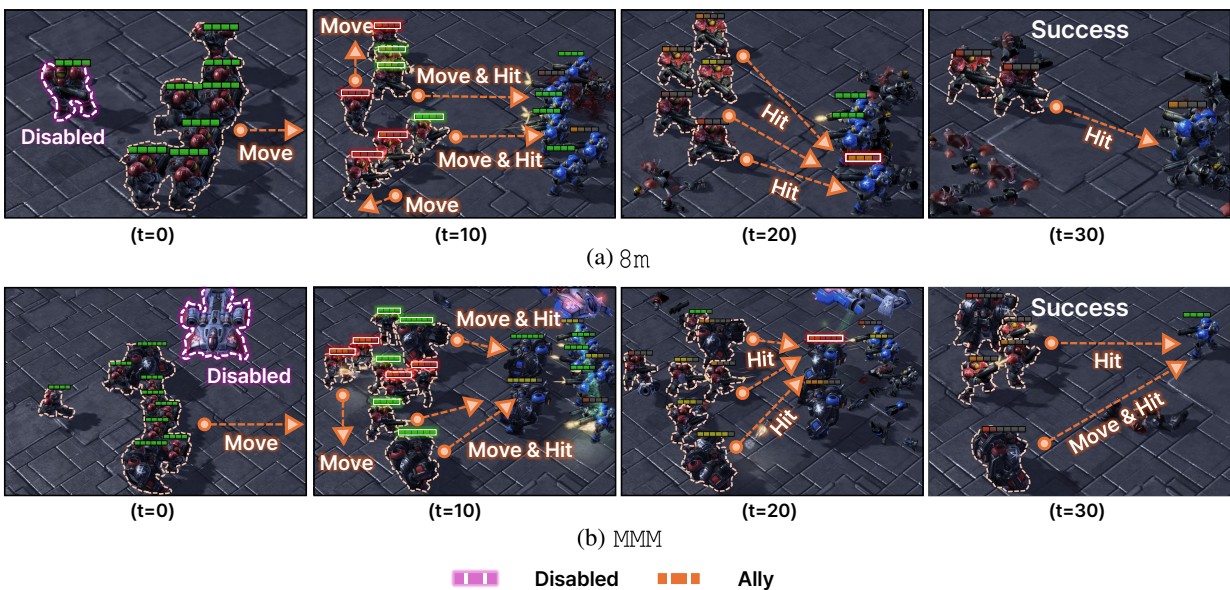

*Figure 15.* Trajectory analysis of IBAL under Dis-1 setting on 8m and MMM scenarios.

In the main paper, we analyzed how agents respond to the proposed interaction-breaking attack that suppresses cross-group coordination. Here, we further examine IBAL under non-parametric perturbations, focusing on Dis-1 in 8m and MMM.

Under Dis-1, one agent becomes disabled and remains stationary, contributing no further actions to cooperation. Despite this failure, IBAL reorganizes the remaining agents into a new coordination pattern in 8m: at $t = 10$, low-health agents fall back while a healthier ally holds the front line; by $t = 20$, the surviving agents concentrate fire on a single target and still complete the task. A similar adaptation appears in MMM. Disabling the medivac removes healing and creates a critical disadvantage, yet IBAL again exhibits coordinated re-formation among the remaining agents: low-health units retreat to preserve survivability while healthier units maintain the front, agents either focus-fire an enemy unit or prioritize the enemy medivac to regain combat advantage and succeed. Overall, these behaviors indicate that IBAL learns resilient group-wise coordination that can be re-formed even when a key contributor is removed, maintaining robustness not only to interaction-breaking attacks but also to non-parametric perturbations such as Dis-$\ell$.

### F.3. Additional Ablation Studies

In this section, we provide additional ablation studies on the masking budget $L$ and the minimum attack probability in the 2s3z and 8m environments, under Dis-1 where the effects of these ablations are most pronounced.

**Masking Budget $L$.** We analyze the effect of the masking budget $L$, defined as a per-agent budget $|D^i| = L$ for the MI-based observation attacker, where $D^i$ denotes the set of masked observation dimensions for agent $i$. For attacked agents $i \in G_1$, we set $L \propto |G_2|$ because $G_1$ masks information about $G_2$, and the amount of cross-group information to block increases as the number of agents in $G_2$ grows. Fig. 16 shows results. When $L = 0$, no observation attack is applied. With a small budget, the attacker cannot sufficiently reduce cross-group MI, resulting in weaker interaction breaking and limited robustness gains. In contrast, an overly large budget masks excessive information from observations, making the perturbation too severe and hindering robust policy learning. Overall, performance is maximized at an intermediate masking budget. We find that $L = 8 \times |G_2|$ works best on 2s3z, while $L = 5 \times |G_2|$ is optimal on 8m, which we use as the default setting.

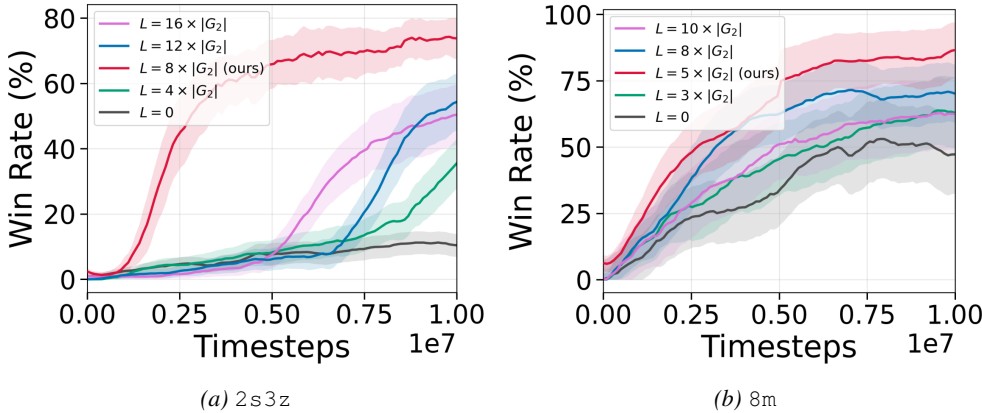

*(a)* 2s3z              *(b)* 8m

*Figure 16.* Ablation study for the masking budget $L$.

**Minimum Attack Probability $P_{\text{act}}^{\min}$.** We ablate the minimum attack probability $P_{\text{act}}^{\min}$ in our scheduling scheme, where $P_{\text{act}} \sim \text{Unif}\big(P_{\text{act}}^{\min}, P_{\text{act}}^{\max}\big)$ is sampled during training. In this ablation, we use $K = 1$ for 2s3z and $K = 4$ for 8m; thus, we sweep $P_{\text{act}}^{\min}$ up to the maximum probability for each environment. As shown in Fig. 17, when $P_{\text{act}}^{\min}$ is too small, action perturbations are rarely applied, yielding insufficient interaction breaking; consequently, robustness improves more slowly. In contrast, an overly large $P_{\text{act}}^{\min}$ makes action perturbations excessively frequent, leading to overly disruptive training dynamics and degraded policy learning. Across attacks and scenarios, we find that $P_{\text{act}}^{\min} = 1/K$ performs best on average, and we adopt $1/K$ as the default across all environments, which corresponds to attacking one agent on average.

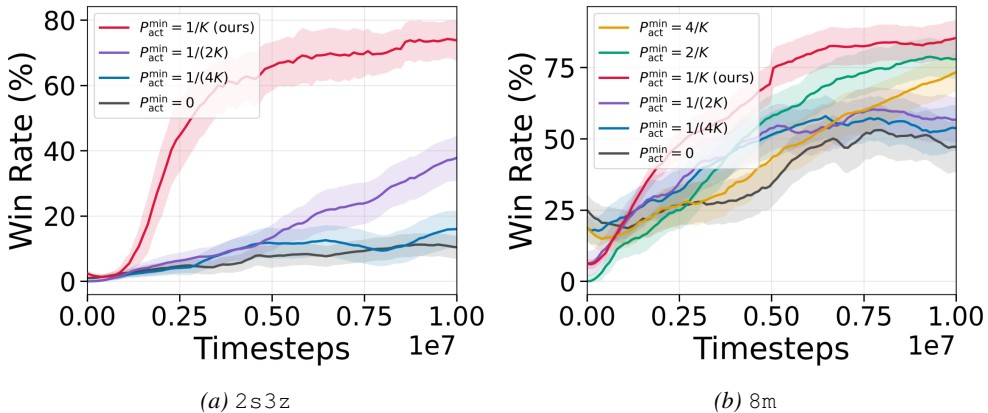

*(a)* 2s3z              *(b)* 8m

*Figure 17.* Ablation study for the minimum attack probability $P_{\text{act}}^{\min}$.

## F.4. Computational Complexity Analysis

We report the total training time required to reach 10M environment steps in SMAC, summarized in Table 14. All times are reported in hours and minutes (h, m). Compared to Vanilla QMIX, our method incurs additional overhead due to mutual-information estimation via observation and action reconstruction models; nevertheless, IBAL is only about $18\%$ slower than QMIX on average. In contrast, methods that require training an RL-based adversarial attacker (ATLA, ROMANCE) or using a Transformer model (WALL) are substantially more expensive, requiring about $91\%$, $80\%$, and $80\%$ more training time than IBAL on average, respectively. Overall, while IBAL is slightly slower than Vanilla QMIX, it achieves consistently stronger robustness with a much lower training-time overhead than ATLA, ROMANCE, and WALL. We also observe that training becomes slower when moving from 3m to MMM, as the number of agents increases and the resulting scalability burden raises the computational cost across all methods.

*Table 14.* Total training time comparison across various SMAC scenarios (h: hour, m: minute).

| Scenario
Model | 3m | 2s3z | 8m | MMM | Mean |
|---|---|---|---|---|---|
| Vanilla QMIX | 36 h 34 m | 42 h 13 m | 52 h 54 m | 57 h 34 m | 47 h 19 m |
| FGSM | 46 h 55 m | 52 h 58 m | 54 h 40 m | 63 h 35 m | 54 h 32 m |
| ATLA | 100 h 52 m | 95 h 47 m | 129 h 12 m | 99 h 04 m | 106 h 14 m |
| ERNIE | 41 h 30 m | 49 h 00 m | 55 h 12 m | 58 h 10 m | 50 h 58 m |
| ROMANCE | 53 h 48 m | 105 h 24 m | 110 h 53 m | 130 h 10 m | 100 h 04 m |
| WALL | 90 h 03 m | 97 h 15 m | 103 h 17 m | 110 h 42 m | 100 h 19 m |
| IBAL (ours) | 48 h 12 m | 49 h 53 m | 59 h 00 m | 65 h 40 m | 55 h 42 m |

