# OpenReview forum: "Interaction-Breaking Adversarial Learning Framework for Robust Multi-Agent Reinforcement Learning"
_ICML.cc/2026/Conference — ICML 2026 regular_

### Official Review · Reviewer_8L32 · 2026-02-27

**Soundness:** 3
**Presentation:** 3
**Significance:** 1
**Originality:** 2
**Overall Recommendation:** 4
**Confidence:** 4

**Summary:**

The paper proposes an robust multi-agent reinforcement learning (MARL) approach based on mutual information minimization. The main idea is to split the agent team into a victim and a functional group and manipulate the victim group (via observation masking and action modification) such that the cross-group information is mutually minimized. The approach, called IBAL (Interaction-Breaking Adversarial Learning), is evaluated on some maps of the old StarCraft II benchmark and shown to be superior to alternative robust MARL methods.

**Compliance With Llm Reviewing Policy:**

Affirmed.

**Final Justification:**

After providing the additional results, I am convinced about its effectiveness. The method is incremental in my view (as discussed above), which is why I only raise my rating to weak accept.

**Key Questions For Authors:**

1. According to the code and configuration files, epsilon-greedy is used for action selection. Has epsilon-greedy also been used for the test/evaluation of the robustness experiments?
2. As far as I understood, the approach does not explicitly incorporate interaction graphs, but masks observations symmetrically based on mutual observability. Given this symmetric relationship, would it be possible to consider the whole interaction graph to define suitable partitions, instead of just sampling them randomly?
3. Figure 9: The approach seems sensitive to the choice of $K$. Apart from tuning it specifically for each domain, is there any heuristic or adaptive approach that could be applied here?

**Limitations:**

Yes

**Strengths And Weaknesses:**

**Originality**

The paper frames its robust MARL approach as an interaction-breaking alternative to prior work, which also manipulates observations and actions. The adversarial Dec-POMDP formulation and the induced robust Dec-POMDP are not novel, given our knowledge about MDPs (where stochastic actions can be viewed as stochastic dynamics without loss of generality) [1] and robust MDPs [2].

The only major difference to prior work, is the optimization objective, which is the mutual information in the paper's case, and the value function in the case of prior works. Given that mutual information has been previously maximized to learn cooperative behavior, as discussed in Section 2 (Related Work), simply minimizing it is expected to lead to performance degradation. Thus, I regard the main contribution as incremental.

**Soundness**

Since the paper is built on known foundations [1,2], the setting of the induced robust Dec-POMDP is sound.

However, I am concerned about the instantiation of the proposed framework: QMIX learns deterministic policies, which act greedy with respect to the decentralized utilities [3]. In the code, epsilon-greedy is used for exploration (but maybe not for testing; see questions below), which does not capture the mixed-strategy concept sufficiently in non-trivial tasks, as all actions are merely treated equally unless an action is the best choice. Breaking the interaction of deterministic policies is easier than stochastic ones, according to the mixed-strategy concept of non-cooperative game theory [4,5,6].

**Presentation**

The paper is mainly well-written and easy to understand. Verifying the theoretical statements is tedious, since all proofs are in the appendix. For better readability and self-containment, I suggest to either provide a proof sketch (if it is important) or move the whole statement to the appendix (if it is less important).

However, I have concerns regarding the framing of the addressed problem:

*“While effective under their targeted perturbation models, these methods often fail to capture breakdowns in the interaction structure underlying coordination. As a result, robustness can remain limited when agents cannot reliably interact or when coordinated attacks disrupt their dependencies, causing substantial performance degradation.”*

As stated above, since the proposed work also just manipulated observations and actions in a similar manner to the prior work, as discussed in Section 2. The only difference is the minimization of the mutual information instead of the learned value function. Thus, the attacks illustrated in Figure 1 can also represent other works if the hint of mutual information is excluded.

**Significance**

The experimental results seem impressive by the numbers. However, the evaluation is conducted on the old/outdated version of the StarCraft II benchmark, which has been criticized for its deterministic dynamics [7,8,9] (fixed initial states and deterministic observations), which makes the policies very easy to disrupt through simple means. Therefore, I do not regard the results as overwhelming.

To improve the significance of the work, I suggest evaluating in more stochastic domains, such as SMACv2 [7] or smaller Dec-POMDP benchmarks, such as Dec-Tiger [8,9,10]. Policies trained in highly stochastic domains are more robust, and therefore harder to break [11].

**Literature**

[1] Sutton and Barto, "Reinforcement Learning: An Introduction"

[2] Wiesemann et al., "Robust Markov Decision Processes", Mathematics of Operations Research, 2013

[3] Rashid et al., "Monotonic Value Function Factorisation for Deep Multi-Agent Reinforcement Learning", JMLR 2020

[4] Littman, "Markov Games as a Framework for Multi-Agent Reinforcement Learning", ICML 1994

[5] Shoham and Leyton-Brown, "Multiagent Systems: Algorithmic, Game-Theoretic, and Logical Foundations", 2008

[6] Albrecht et al., "Multi-Agent Reinforcement Learning: Foundations and Modern Approaches", MIT Press, 2024

[7] Ellis et al, "SMACv2: An Improved Benchmark for Cooperative Multi-Agent Reinforcement Learning", NeurIPS Benchmarks 2023

[8] Lyu et al., "A Deeper Understanding of State-Based Critics in Multi-Agent Reinforcement Learning", AAAI 2022

[9] Phan et al., "Attention-Based Recurrence for Multi-Agent Reinforcement Learning under Stochastic Partial Observability", ICML 2023

[10] Oliehoek and Amato, "A Concise Introduction to Dec-POMDPs", Springer 2016

[11] Eysenbach et al., "Maximum Entropy RL (Provably) Solves Some Robust RL Problems", ICLR 2022

---

> ### Author Rebuttal · Authors · 2026-03-31
>
> ## Reviewer 8L32
>
> We sincerely thank the reviewer for the constructive feedback. We would like to address the reviewer’s concerns as follows. We will incorporate all of the points addressed in our response into the revised manuscript.
>
> **Weakness (Novelty of IBAL).** We thank the reviewer for the thoughtful comment. We would like to clarify that IBAL is not simply an extension of existing robust MARL methods, but introduces two core novelties. First, IBAL proposes **a fundamentally new attack concept.** While prior robust MARL methods mainly design attacks using value-related signals, IBAL introduces interaction-breaking, an attack mechanism based on minimizing inter-agent mutual information. To the best of our knowledge, this is the first work to explicitly formulate interaction disruption as the basis of adversarial training in MARL. We believe this is an important conceptual contribution, as it shifts the focus from value degradation to the disruption of inter-agent coordination itself.
>
> Second, **the mechanism used to realize this idea is also fundamentally different from existing methods.** Prior robust MARL approaches typically construct attacks by directly minimizing value-related quantities, often through action perturbations. In contrast, IBAL introduces a new observation-space attack mechanism for minimizing mutual information, enabled by additional decomposition and a zero-forcing masking design for joint attacks. Therefore, IBAL is not merely a modified objective built on prior methods, but a distinct attack framework with a new tool for interaction-level robustness. We will revise the manuscript to make these novelties more explicit.
>
> **Weakness (Additional benchmarks).** We appreciate the reviewer’s valuable suggestion. To further validate IBAL beyond SMAC, we conducted additional experiments on two environments suggested by the reviewer: **Level-Based Foraging (LBF)** and **SMACv2**, with the former also suggested by Reviewer wy4t. For LBF, we evaluated Foraging-20x20-np-6f-s3 with $n = 5$ and $10$ agents. For SMACv2, we considered three scenarios: terran_5_v_5, zerg_5_v_5, and protoss_5_v_5. In accordance with the reviewer guidelines, we report the comparison results in Table R1 in our response to Reviewer wy4t, and kindly refer the reviewer to that table for details. Across both environments, IBAL consistently achieves the highest return in all settings. While baseline performance degrades substantially under interaction-breaking attacks, IBAL remains strong against most attack types and also shows improved natural performance in smaller Dec-POMDPs such as LBF, as well as in highly stochastic environments such as SMACv2. These results indicate that **IBAL generalizes well to diverse MARL environments beyond SMAC.**
>
> **Weakness (Location of the proof).** We appreciate the reviewer’s suggestion and will revise the manuscript to make the proof more self-contained and easier to locate.
>
> **Question 1 (Training/Test policy).** We thank the reviewer for the question. Epsilon-greedy exploration is used only during training, while all evaluations are conducted with a greedy policy. This evaluation protocol is fully consistent with prior work. In robust MARL, the key question at test time is not whether a deterministically trained policy remains robust when attacks are applied afterward, but **whether the policy learned under adversarial training can generalize effectively to unseen attacks.** From this perspective, the level of exploration during training is not a central factor in the evaluation itself.
>
> **Question 2 (Advanced grouping strategies).** We appreciate the reviewer’s valuable suggestion. In our current implementation, we use random group partitioning to expose the model to diverse group configurations. As the reviewer suggested, a more advanced approach would be to model agent relationships as an MI-based interaction graph and prioritize groups whose disruption would most severely impair cooperation. While we did not explore this direction in the current work due to the complexity of our method, we agree that it is a valuable avenue for future research.
>
> **Question 3 (Sensitivity to $K$).** As the reviewer pointed out, the maximum group size $K$ controls the strength of the attack, so setting it too large or too small can harm robustness. Therefore, choosing an appropriate value is important. In our experiments, we found that setting **$K$ roughly in proportion to the total number of agents**, approximately half of them, works stably across environments. Accordingly, for a new environment, we expect stable performance when initializing $K$ to about half the number of agents and then adjusting it as needed.

---

> > ### Author Rebuttal · Reviewer_8L32 · 2026-04-03
> >
> > I am happy with the answers to Q2 and Q3. However, I am not satisfied with the answer to Q1 because breaking a deterministic system is generally easier than a stochastic system (according to Game/Minimax theory).
> >
> > Regarding novelty, my opinion remains the same: past work has shown that maximizing mutual information improves performance. Thus, minimizing it should deteriorate it, which is why I am not intrigued/surprised by the finding.
> >
> > I appreciate comments about the additional benchmark results, but without any numbers, I am not convinced scientifically.

---

> > > ### Author Response · Authors · 2026-04-06
> > >
> > > We thank the reviewer for the additional comments. First, regarding the benchmark experiments, as noted above, due to the space limitation and following the reviewer guidelines, we reported **the numerical results (Table R1) in our response to Reviewer wy4t**, who raised a similar question. We would be grateful if the reviewer could kindly refer to that table.
> > >
> > > Second, to address Q1 more directly regarding stochasticity, we provide in Table R5 the results of IBAL when both training and testing are conducted with $\epsilon$-greedy policies. As shown in the results, even with **stochasticity at test time, the method remains sufficiently robust in terms of both attack effectiveness and defense performance.**
> > >
> > > Finally, regarding novelty, although it is known that increasing mutual information can improve performance, and it may therefore seem intuitive that decreasing it could harm performance, we would gently argue that this does not necessarily make the idea of attacking mutual information trivial. As discussed in our rebuttal, our work is, to the best of our knowledge, the first to explicitly formulate mutual information as an attack target, and we believe that this perspective, together with its realization in the MARL setting, offers meaningful novelty.
> > >
> > > **Table R5.** Average test win rate under test-time $\epsilon$-greedy evaluation ($\epsilon$=5%) for the results reported in manuscript Figure 5.
> > >
> > > |Scenario|Model|Nat.|Rand.|FGSM|EGA|Wolf.|Ours|
> > > |---|---|---|---|---|---|---|---|
> > > |3m|VanillaQMIX|98.8±1.3|41.8±7.0|78.4±5.6|65.0±16.2|43.1±8.1|0.0±0.0|
> > > ||Rand.|99.0±1.0|42.5±6.5|86.3±5.9|51.0±29.0|54.2±10.1|0.0±0.0|
> > > ||FGSM|96.7±4.2|53.3±2.1|**99.3±1.2**|49.0±9.6|56.2±16.5|3.3±5.8|
> > > ||ATLA|**99.3±1.2**|55.3±3.1|85.7±1.5|38.5±6.5|51.0±9.6|0.0±0.0|
> > > ||ERNIE|98.7±0.6|42.0±5.2|80.7±5.1|49.0±20.8|39.6±17.2|0.0±0.0|
> > > ||ROMANCE|93.4±13.7|55.4±4.7|63.0±6.9|84.8±11.2|43.1±21.6|0.0±0.0|
> > > ||WALL|97.8±2.9|43.2±4.8|64.8±16.1|80.6±14.2|**77.5±7.1**|1.0±2.2|
> > > ||IBAL(ours)|98.0±1.0|**58.7±7.0**|96.0±4.4|**90.6±5.9**|67.7±3.2|**75.1±5.6**|
> > > |3s_vs_3z|VanillaQMIX|97.3±2.1|75.2±9.2|75.6±20.8|83.1±4.1|67.5±6.1|3.0±4.5|
> > > ||Rand.|98.5±2.1|82.8±4.0|85.5±0.7|85.9±4.4|90.2±6.6|10.0±14.1|
> > > ||FGSM|97.3±1.2|81.3±1.2|**97.0±2.6**|68.8±16.2|90.6±5.4|3.3±5.8|
> > > ||ATLA|99.0±1.4|83.0±12.2|88.0±1.0|70.8±26.0|94.8±3.6|6.7±7.6|
> > > ||ERNIE|97.0±2.6|83.0±4.4|76.3±11.2|85.4±12.6|95.8±4.8|5.0±8.7|
> > > ||ROMANCE|98.5±2.1|81.0±1.6|91.2±3.5|92.4±1.4|95.4±1.4|26.0±15.2|
> > > ||WALL|98.6±1.2|80.4±10.4|85.4±7.8|92.5±2.6|97.8±1.7|17.0±12.0|
> > > ||IBAL(ours)|**99.8±0.5**|**88.5±0.7**|93.5±3.5|**95.6±3.6**|**99.4±1.4**|**85.0±15.0**|
> > > |2s3z|VanillaQMIX|98.2±0.8|82.1±6.2|63.8±4.2|58.1±5.2|33.1±14.4|2.0±2.7|
> > > ||Rand.|98.7±1.2|83.4±6.4|84.3±1.5|65.6±8.3|62.5±5.4|0.0±0.0|
> > > ||FGSM|99.0±1.7|84.3±7.6|**96.7±0.6**|65.6±0.0|65.6±10.8|10.0±5.0|
> > > ||ATLA|97.7±1.5|78.0±10.2|85.3±4.0|57.3±11.0|43.8±11.3|5.0±0.0|
> > > ||ERNIE|98.7±0.6|82.7±4.9|77.3±8.4|62.5±5.4|58.3±10.1|6.7±2.9|
> > > ||ROMANCE|96.8±1.8|85.6±4.3|68.0±10.1|83.8±6.8|71.9±9.9|7.0±6.7|
> > > ||WALL|98.2±2.2|89.2±4.3|85.6±7.1|86.2±7.2|87.5±11.3|11.0±5.5|
> > > ||IBAL(ours)|**99.2±1.0**|**91.6±3.5**|88.3±7.4|**86.8±5.3**|**89.0±5.9**|**79.6±4.5**|
> > > |8m|VanillaQMIX|99.2±1.1|87.2±4.0|92.4±5.4|89.4±9.8|20.3±2.2|5.0±8.7|
> > > ||Rand.|98.0±2.0|86.2±4.8|93.7±2.3|74.0±13.0|29.2±3.6|6.7±2.9|
> > > ||FGSM|99.0±1.0|90.7±1.5|**97.3±0.6**|83.8±5.4|41.7±4.8|15.0±8.7|
> > > ||ATLA|96.0±2.7|79.0±7.8|93.3±2.5|85.4±12.6|45.8±7.9|8.3±7.6|
> > > ||ERNIE|98.7±1.5|81.7±4.9|91.0±5.0|77.1±4.8|33.3±10.1|3.3±2.9|
> > > ||ROMANCE|99.0±1.7|94.3±4.5|93.0±5.1|90.0±3.6|48.1±16.0|3.0±2.7|
> > > ||WALL|99.2±1.3|90.3±2.7|91.3±6.2|92.4±1.7|86.5±7.0|4.2±5.9|
> > > ||IBAL(ours)|**99.3±1.2**|**94.6±3.8**|95.6±4.0|**95.8±1.8**|**86.6±2.9**|**73.3±12.6**|
> > > |1c3s5z|VanillaQMIX|99.0±0.7|88.3±3.0|84.6±3.0|79.4±1.7|41.9±8.4|14.0±3.7|
> > > ||Rand.|98.7±0.6|86.3±3.3|82.0±2.7|83.3±6.5|41.7±3.6|10.4±5.6|
> > > ||FGSM|99.0±1.7|85.3±2.1|93.0±1.0|86.5±6.5|77.5±3.1|10.0±4.5|
> > > ||ATLA|98.0±0.0|86.7±1.2|84.3±3.1|82.3±1.8|89.6±15.4|12.9±6.2|
> > > ||ERNIE|98.0±2.6|87.5±3.5|88.5±2.1|79.7±6.6|58.3±51.4|8.3±7.6|
> > > ||ROMANCE|97.8±1.8|84.0±4.4|89.6±4.2|83.8±8.1|85.6±2.8|13.5±2.1|
> > > ||WALL|99.0±0.7|95.0±1.6|97.0±1.4|**96.9±3.8**|88.6±3.2|16.7±3.3|
> > > ||IBAL(ours)|**99.7±0.6**|**98.0±0.8**|**97.5±2.4**|92.2±6.5|**91.4±6.9**|**91.2±2.4**|
> > > |MMM|VanillaQMIX|99.6±0.6|86.4±2.8|83.8±2.2|71.2±14.7|27.2±13.0|1.0±2.2|
> > > ||Rand.|98.7±1.5|87.9±1.8|87.0±3.6|78.1±8.3|64.2±8.3|0.0±0.0|
> > > ||FGSM|**99.7±0.6**|88.0±3.5|89.0±1.7|79.2±7.2|62.7±12.6|1.7±2.9|
> > > ||ATLA|99.0±1.0|86.3±3.2|86.0±4.0|72.9±11.0|61.9±8.8|3.3±5.8|
> > > ||ERNIE|99.0±1.7|86.3±0.6|88.3±1.2|72.9±6.5|61.9±4.6|3.3±5.8|
> > > ||ROMANCE|95.8±7.7|86.8±4.5|85.6±8.2|89.4±8.2|52.0±4.9|1.0±2.2|
> > > ||WALL|92.2±10.3|91.4±12.4|84.6±8.4|91.9±7.2|90.0±12.4|0.0±0.0|
> > > ||IBAL(ours)|98.1±1.7|**96.6±3.2**|**96.4±1.9**|**95.7±1.8**|**92.7±1.9**|**87.5±10.6**|

---

### Official Review · Reviewer_haoF · 2026-03-07

**Soundness:** 3
**Presentation:** 3
**Significance:** 3
**Originality:** 3
**Overall Recommendation:** 4
**Confidence:** 4

**Summary:**

This paper proposes a robust multi-agent reinforcement learning framework called IBAL, designed to address the vulnerability of cooperative policies when external disturbances disrupt inter-agent interactions. From an information-theoretic perspective, IBAL randomly partitions agents into groups and uses MI to quantify cross-group influence. Based on this measurement, it constructs adversarial attacks capable of masking critical observation dimensions and perturbing coordinated actions, thereby simulating scenarios where the cooperative structure is partially or completely compromised. By performing adversarial training within the induced Dec-POMDP process, IBAL compels agents to acquire adaptive coordination capabilities, even in situations where they cannot reliably observe their teammates or when collaboration breaks down. Experimental results demonstrate that IBAL not only outperforms existing baselines in defending against various adversarial attacks across multiple complex SMAC scenarios, but also exhibits stronger robustness and generalization under non-parametric disturbances such as agent dropout or reduced health.

**Compliance With Llm Reviewing Policy:**

Affirmed.

**Final Justification:**

As mentioned in my "Rebuttal Acknowledgement", I still have the concern that the robustness boundaries of IBAL remain unclear when applied to tasks with complex, phase-structured objectives.

Thus, I prefer to keep my current score.

**Key Questions For Authors:**

IBAL’s attack mechanism minimizes instantaneous MI at every time step . However, in tasks such as Overcooked or multi‑stage logic‑dependent environments, successful cooperation often hinges on the accurate exchange of information at a specific temporal checkpoint. Does this uniformly distributed “interaction noise” training truly prevent agents from suffering global tactical failures when critical‑moment information is lost? Since IBAL’s reconstruction model depends on the historical trajectory $\tau_t$​, if a key cooperative signal occurred hundreds of steps earlier, do the current reconstruction models $p_\theta$​ and $p_\phi$​ possess sufficient long‑horizon memory to recognize and disrupt interaction patterns based on such past agreements?


In non‑monotonic tasks, the individually optimal behaviors typically decouple from the globally optimal ones. When IBAL’s action attack selects the MI‑minimizing action $\tilde{a}_t^{\text{min}, G_1}$, such perturbations may cause severe gradient‑direction distortions under non‑monotonic credit assignment. How can IBAL ensure training stability and robustness when applied to algorithms without monotonicity guarantees, such as MAPPO or other non‑decomposition methods based on reward shaping or Actor–Critic updates?

In environments requiring risk‑sensitive collaboration, agents often need to exchange strong, high‑MI signals to achieve high rewards. Since IBAL explicitly minimizes MI, does the adversarial pressure push agents toward overly conservative, independent strategies, potentially collapsing to a locally optimal but non‑cooperative solution?

If the reconstruction model $p_\theta$​ is biased in early training due to insufficient samples, could this bias be amplified through adversarial learning, ultimately steering agents toward incorrect defensive behaviors? In high‑dimensional observation spaces, does the MI‑based dimension selection suffer from the curse of dimensionality, potentially degrading the accuracy of the attack and causing computation to become prohibitively expensive in environments more complex than SMAC?

**Limitations:**

Please refer to the weaknesses.

**Strengths And Weaknesses:**

The core of IBAL’s attack mechanism lies in minimizing instantaneous mutual information. In the paper, both the observation attack $f_{adv}$​ and the action attack $\pi_{adv}$​ are applied at every time step $t$. Although SMAC tasks are complex, many micro‑level actions have immediate effects. In contrast, in cooperative tasks involving multi‑stage objectives (e.g., “pick up the key before opening the door”), coordination failure may occur at a critical temporal checkpoint. IBAL’s uniformly distributed, per‑step random attacks may merely increase training noise rather than accurately simulating long‑horizon task failures caused by losing coordination at a pivotal moment.

The paper estimates mutual information using reconstruction models based on historical trajectories $\tau_t$​. In environments with very long temporal dependencies, the current observation may embed interaction information from far earlier time steps. Whether IBAL’s current reconstruction models $p_\theta$​ and $p_\phi$​ can capture nonlinear interaction patterns spanning hundreds of steps is not validated in the paper. If the adversary can only disrupt moment‑to‑moment interactions, IBAL’s robustness may degrade in tasks that rely on “past agreements.”

IBAL is primarily implemented on top of QMIX, which embeds it deeply within the Individual–Global–Max principle and the associated monotonicity constraints. Theorem 4.2 in the paper proves the existence of an equivalent induced Dec‑POMDP under adversarial attacks. However, when translating this result into an actual algorithmic implementation, the proof relies on QMIX’s monotonic mixing network to maintain stable credit assignment. In non‑monotonic environments (e.g., tasks with reward traps) or when using non–value‑decomposition algorithms such as Actor–Critic–based MAPPO, an adversary that minimizes mutual information may cause large oscillations in the policy‑gradient direction rather than the stable robustness improvements reported in the paper.

In non‑monotonic tasks, the optimal joint action cannot be expressed as a simple combination of individually optimal actions. IBAL’s action‑level attack disrupts coordination by selecting the MI‑minimizing action a~tmin, $\tilde{a}_t^{\text{min}, G_1}$​​. Under QMIX, monotonicity ensures that such disruptions typically manifest as conservative behaviors such as “retreating.” However, in more complex social dilemmas or zero‑sum settings, this MI‑based disruption logic may inadvertently harm the agents’ ability to learn risk‑sensitive collaboration, because it may misinterpret complex collaborative signals as adversarial interference and consequently drive the agents toward overly conservative, independent strategies.

---

> ### Author Rebuttal · Authors · 2026-03-31
>
> ## Reviewer haoF
>
> We sincerely thank the reviewer for the constructive feedback. We would like to address the reviewer’s concerns as follows. We will incorporate all of the points addressed in our response into the revised manuscript.
>
> **Weakness 1& Question 1 (Critical-time-step selection).** As the reviewer pointed out, robustness also requires handling critical moments. In principle, IBAL could be extended to select critical time steps that maximally reduce mutual information, similar to other attack-based methods. However, to avoid additional computational complexity, we instead adopt an adaptive attack schedule that starts with a low attack frequency and gradually increases it over training. **This allows the policy to experience interaction disruption more frequently and improves the chance of covering critical moments.** We view this as a key design choice of IBAL, and our ablation study shows that it contributes to improved robustness.
>
> **Weakness 2 & Question 2 (Long-term influence).** In many MARL environments, agents primarily influence one another through immediate interactions, so minimizing instantaneous mutual information (MI) is often sufficient in practice. As the reviewer pointed out, however, there may also be settings in which a current action affects information far into the future. In such cases, IBAL could be extended in a relatively straightforward manner, for example by incorporating a Transformer-based estimator of future MI or by minimizing the sum of MI over multiple future steps. While these extensions are beyond the scope of the current paper, we believe they represent valuable directions for future research.
>
> **Weakness 3 & Question 3 (Applicability beyond QMIX).** We appreciate the reviewer’s question. Although we use QMIX as the baseline in this paper, IBAL does not rely on value information for the attack, unlike many existing attack-based MARL methods. As a result, **IBAL is conceptually independent of the IGM property of QMIX** and is, in principle, compatible with a broad range of MARL algorithms. To illustrate this more directly, following the reviewer’s suggestion, we additionally apply IBAL to MAPPO and report the results in Table R3. The results show that IBAL also maintains strong performance and robustness with MAPPO, further supporting that its effectiveness is not tied to a specific MARL method.
>
> **Table R3.** Comparison of attack methods applied to MAPPO on SMAC (Win rate)
>
> |Attack|Model|3s_vs_3z|2s3z|8m|
> |---|---|---:|---:|---:|
> |Natural|MAPPO|98.0±2.8|97.9±2.7|98.5±0.7|
> ||MAPPO+Rand-Obs|99.0±1.4|98.4±2.2|97.4±0.8|
> ||MAPPO+Rand-Act|97.4±0.8|98.0±2.8|95.9±1.3|
> ||MAPPO+FGSM|**99.0±1.4**|99.5±0.6|99.0±1.4|
> ||MAPPO+ATLA|98.5±2.1|99.5±0.6|**100.0±0.0**|
> ||MAPPO+IBAL|98.5±2.1|**100.0±0.0**|98.3±2.5|
> |Rand.|MAPPO|75.0±4.4|81.8±8.1|76.6±2.2|
> ||MAPPO+Rand-Obs|74.5±5.0|84.4±4.4|97.5±3.5|
> ||MAPPO+Rand-Act|82.8±6.6|89.1±2.2|98.1±0.1|
> ||MAPPO+FGSM|95.0±7.1|95.9±1.2|95.3±2.2|
> ||MAPPO+ATLA|82.8±6.6|95.3±2.2|92.2±2.2|
> ||MAPPO+IBAL|**99.0±1.4**|**96.9±0.1**|**98.4±2.2**|
> |Ours|MAPPO|4.7±2.2|11.3±1.8|4.7±6.6|
> ||MAPPO+Rand-Obs|1.6±2.2|16.8±6.0|25.0±4.4|
> ||MAPPO+Rand-Act|4.7±2.2|20.9±1.3|18.6±4.6|
> ||MAPPO+FGSM|17.2±11.0|45.3±2.2|23.4±6.6|
> ||MAPPO+ATLA|0.0±0.0|26.6±11.0|18.8±17.7|
> ||MAPPO+IBAL|**97.4±0.8**|**95.3±2.2**|**95.0±2.8**|
>
> **Weakness 4 & Question 4 (Concern about overly strong attacks).** The risk of over-conservatism due to excessive attack strength or early training bias is a common concern in adversarial training, and our MI-based approach does not inherently exacerbate this issue. **To mitigate it, we incorporate several safeguards.** First, training includes both attacked and normal conditions, which prevents the policy from becoming overly biased toward adversarial cases. Second, we use attack-probability scheduling to avoid overly strong attacks in the early stage of training and increase the attack intensity only as training progresses. These design choices allow **IBAL to improve robustness without inducing excessive conservatism.**
>
> **Question 5 (Early bias due to the curse of dimensionality).** This concern is closely related to our response to Question 4. While imperfect MI estimation may produce somewhat biased attacks, our safeguards prevent such errors from causing the policy itself to become overly biased in practice. Since the agent is trained under a range of attacks that remains sufficiently manageable, we do not expect estimation errors to introduce excessive bias into the learned policy. Regarding the curse of dimensionality, our MI estimation operates on individual observation dimensions rather than the joint space, which significantly reduces computational complexity. As confirmed in Appendix F.4, overhead remains moderate even with increasing agent count, indicating that the approach scales reasonably to more complex settings.

---

> > ### Author Rebuttal · Reviewer_haoF · 2026-04-01
> >
> > The authors argue that “adaptive attack frequency” is used to cover critical moments. While this may represent a practical engineering trade-off, from a theoretical perspective, it still amounts to random coverage and does not fundamentally address the problem of identifying and precisely targeting “critical nodes” in cooperative tasks. As a result, the robustness boundaries of IBAL remain unclear when applied to tasks with complex, phase-structured objectives, such as Overcooked.
> >
> > Thus, I prefer to keep my score.

---

> > > ### Author Response · Authors · 2026-04-05
> > >
> > > We thank the reviewer for the insightful comment. We would like to clarify more concretely that the adaptive scheduling in our proposed IBAL can effectively handle critical moments.
> > >
> > > To respond more directly to this point, we conducted an additional experiment on SMAC 8m in Table R4. In particular, we compare **IBAL (original)**, with the proposed adaptive probability scheduling, against **IBAL (critical attack)**, which applies the critical-moment attack strategy proposed in ROMANCE [R.1]. While the original critical attack selects value-minimizing attack timing, in our setting we adapt it by attacking the time steps that minimize mutual information.
> > >
> > > The results show that **the proposed IBAL (original) achieves better performance than IBAL (critical attack), even when the attacks are explicitly focused on critical moments.** As discussed in our rebuttal, we believe this is because the adaptive scheduling naturally increases the attack frequency over time and is therefore able to cover critical moments as well. In addition, **a strategy that mainly targets such critical moments appears to be more vulnerable to random attacks or unseen attack strategies**, which further highlights the advantage of the proposed method. At the same time, we also agree with the reviewer that in highly cooperative environments such as **Overcooked**, where coordination is particularly important, explicitly targeting carefully selected critical moments could potentially be even more effective. Nevertheless, we would like to note that such an extension can be incorporated into our framework in a straightforward manner, as illustrated by the experiment above.
> > >
> > > **Table R4.** Comparison of action attack scheduling strategies on SMAC-8m
> > >
> > > |Attack|Model|Win rate|
> > > |---|---|---:|
> > > |Natural|Vanilla QMIX|92.8±1.9|
> > > ||ROMANCE|98.1±1.0|
> > > ||IBAL (critical attack)|95.8±7.2|
> > > ||IBAL (original)|**99.3±1.1**|
> > > |EGA attack (proposed in R.1) |Vanilla QMIX|70.8±16.0|
> > > ||ROMANCE|90.0±3.8|
> > > ||IBAL (critical attack)|95.8±0.9|
> > > ||IBAL (original)|**97.6±1.2**|
> > > |IBAL attack (critical attack)|Vanilla QMIX|0.0±0.0|
> > > ||ROMANCE|1.3±1.7|228.3±17.6|
> > > ||IBAL (critical attack)|64.5±2.6|
> > > ||IBAL (original)|**69.8±3.6**|
> > > |IBAL attack (original)|Vanilla QMIX|7.3±3.6|
> > > ||ROMANCE|11.0±11.0|
> > > ||IBAL (critical attack)|66.2±13.6|
> > > ||IBAL (original)|**88.4±3.3**|
> > >
> > > [R.1] Yuan et al., "Robust Multi-Agent Coordination via Evolutionary Generation of Auxiliary Adversarial Attackers." AAAI, 2023.

---

### Official Review · Reviewer_R4Y9 · 2026-03-10

**Soundness:** 3
**Presentation:** 3
**Significance:** 4
**Originality:** 3
**Overall Recommendation:** 5
**Confidence:** 4

**Summary:**

The paper proposes IBAL (Interaction-Breaking Adversarial Learning), a framework that reframes robust multi-agent reinforcement learning by focusing on the fragility of coordination against interaction-severing attacks. This research introduces an Interaction-Breaking Attack, a structure composed of Mutual Information (MI)-guided observation masking and action perturbations to explicitly suppress cross-group influence and disrupt coordination. The authors claim this approach outperforms state-of-the-art baselines on StarCraft II (SMAC) benchmarks, particularly exhibiting strong generalization in scenarios where agents are missing or disabled.

**Compliance With Llm Reviewing Policy:**

Affirmed.

**Final Justification:**

I found this paper strong overall, and the rebuttal further increased my confidence in it.

The main strengths are its clear problem formulation, solid technical grounding, and strong empirical evaluation. In particular, I think the paper makes a meaningful contribution by shifting the focus of robust MARL from standard noise-based attacks to interaction-severing attacks, which is a genuinely interesting and useful perspective. The experiments are also thorough and support the claim that IBAL provides a real robustness advantage.

My initial questions were mainly about the exact framing of the contribution, the choice of zero-masking versus more realistic perturbations, and the improvement in the Natural setting. The rebuttal addressed these points well. I especially appreciated the additional experiments comparing zero-masking with Gaussian noise and FGSM perturbations, since they directly strengthened the justification for the proposed attack design. The explanation for the Natural-setting gains was also reasonable and better supported after rebuttal.

So overall, the rebuttal reinforced my original positive view rather than changing it. I believe the paper is technically solid, well evaluated, and likely to be useful to others working on robust multi-agent coordination. My final recommendation is accept.

**Key Questions For Authors:**

Q1: It is somewhat ambiguous whether the contribution lies in proposing the problem of 'interaction breaking' itself or in the specific MI-based method used to address it. Could the authors explicitly state if there are prior works on this specific topic?

Q2: The observation attack primarily employs a 'zero-forcing' mask to block information. However, real-world sensor failures or adversarial attacks often involve noisy or misleading signals rather than pure silence. Did the authors investigate whether replacing masked values with Gaussian noise or adversarial perturbations would yield different robustness characteristics compared to zero-masking?

Q3: It is noted that IBAL improves performance even in the 'Natural' (no-attack) setting compared to Vanilla QMIX. Could the authors provide a more in-depth explanation as to why this improvement occurs in the absence of perturbations?

**Limitations:**

yes

**Strengths And Weaknesses:**

**Soundness**

The soundness of this work is solid, supported by the explicit mathematical formulation of the Joint-Adversarial Dec-POMDP and Theorem 4.2, which prevents the method from being a "black box." Furthermore, the experimental evaluation is comprehensive, comparing against strong, state-of-the-art baselines across diverse scenarios, including non-parametric perturbations. This effectively demonstrates that IBAL offers a genuine robustness advantage over existing solutions.

**Presentation**

The submission is clearly written and well-structured, effectively positioning the work against state-of-the-art robust MARL baselines. Key mathematical formulations, such as the JA-Dec-POMDP and MI-based interaction breaking, are explicitly defined and logically presented. Furthermore, the experimental analysis provides deep insight, using trajectory visualizations to demonstrate specifically how agents adapt coordination under attack. While the overall narrative is compelling, the dense notation regarding MI estimation details could be slightly streamlined to further improve accessibility.

**Significance**
The significance of this work is excellent. The paper addresses a critical gap in robust MARL by shifting the focus from simple observation noise to interaction-severing attacks. The proposed IBAL framework advances the field by leveraging Mutual Information to explicitly model and mitigate coordination failure. Crucially, the theoretical equivalence established in Theorem 4.2 offers a mathematically grounded approach to robust training, while the strong empirical results on non-parametric perturbations demonstrate broad practical utility that will likely influence future research on resilient multi-agent coordination.

**Originality**

The originality of this work is good. The proposed IBAL framework introduces a novel perspective by framing robust MARL as a defense against interaction-severing attacks via Mutual Information (MI) minimization. Unlike prior adversarial MARL methods that predominantly focus on value-minimizing attacks or simple noise injection, this work uniquely targets the specific dependencies and coordination structures between agents. However, In the Related Work section, the authors should list relevant literature regarding 'breaking' interactions between agents, or explicitly state that this is the first study to explore this specific direction.

---

> ### Author Rebuttal · Authors · 2026-03-31
>
> ## Reviewer R4Y9
>
> We sincerely thank the reviewer for the thoughtful and in-depth questions. We would like to address the reviewer’s concerns as follows. We will incorporate all of the points addressed in our response into the revised manuscript.
>
> **Weakness 1 & Question 1 (Related Work \& Novelty).** We appreciate the reviewer’s important comment. **To the best of our knowledge, our work is the first to leverage the disruption of inter-agent interactions as an attack mechanism**, and we agree that this point is important enough to be stated explicitly. Accordingly, we will clarify this more clearly in the revised manuscript. In this sense, we believe our contribution is meaningful in two respects: first, we introduce the concept of improving robustness by exploiting inter-agent interactions rather than value-based signals, and second, we design the attack components necessary to instantiate this idea in practice. We will revise the paper to emphasize these contributions more clearly.
>
> **Question 2 (Additional observation noise).** We initially considered zero-masking because it most directly minimizes mutual information. Following the reviewer’s suggestion, however, we additionally compared our method with more realistic perturbations, including Gaussian noise and adversarial observation perturbation proposed by FGSM, and report the results in Table R2. The results show that our proposed zero-masking achieves the strongest robustness. We believe this is because **the proposed zero-masking completely removes cross-group information**, thereby directly minimizing observation-level mutual information and inducing a stronger attack. In contrast, noise-based perturbations add misleading signals, but still preserve more mutual information than our method, making them less effective at disrupting inter-agent interactions.
>
> **Table R2.** Win rate of IBAL under diverse observation attacks.
>
> | Map | Attack | Win rate |
> | --- | --- | --- |
> | 8m | Ours (Zero-masking) | **88.4±3.3** |
> |  | Ours (Gaussian noise) | 78.1±13.3 |
> |  | Ours (FGSM perturbation) | 38.5±7.4 |
> | MMM | Ours (Zero-masking) | **88.7±7.2** |
> |  | Ours (Gaussian noise) | 71.9±22.1 |
> |  | Ours (FGSM perturbation) | 77.6±3.6 |
>
> **Question 3 (Performance improvement in the 'Natural' setting).** We appreciate the reviewer’s question. While SMAC is largely deterministic, its dynamics still include some stochasticity, which can occasionally cause QMIX to fail at test time. In contrast, IBAL, like other robust training methods, is trained under a broader range of perturbed conditions induced by attacks. We believe this improved exposure to uncertainty helps the policy generalize slightly better, leading to a modest performance gain even in the 'Natural' setting. This tendency is even clearer in SMACv2, whose agent composition varies across episodes and thus exhibits much greater stochasticity. As shown in Table R1 of our response to Reviewer wy4t, added following the suggestions of both Reviewer wy4t and Reviewer 8L32, IBAL achieves substantially better performance even in the natural setting, providing additional support for this explanation.

---

> > ### Author Rebuttal · Reviewer_R4Y9 · 2026-04-01
> >
> > I thank the authors for the clear and helpful rebuttal. My main concerns have been adequately addressed.
> >
> > The clarification of the paper’s contribution is useful, and I especially appreciate the additional experiments comparing zero-masking against Gaussian noise and FGSM-style perturbations. These results directly strengthen the justification for the proposed attack design. I also find the explanation for the improvement in the Natural setting reasonable, with the additional referenced evidence making the claim more convincing.
> >
> > Overall, the rebuttal strengthens my confidence in the paper. Based on these responses, I consider my concerns resolved and have updated my score **from weak accept to accept**.

---

> > > ### Author Response · Authors · 2026-04-05
> > >
> > > We sincerely thank the reviewer for the thoughtful feedback and for raising the assessment of our work. The reviewer’s comments were highly valuable in improving the clarity of our paper. We will reflect the points discussed during the review in the revised manuscript. We greatly appreciate the reviewer’s constructive feedback.

---

### Official Review · Reviewer_wy4t · 2026-03-19

**Soundness:** 3
**Presentation:** 3
**Significance:** 3
**Originality:** 2
**Overall Recommendation:** 5
**Confidence:** 3

**Summary:**

The paper proposes the IBAL framework to address the fragility of cooperative MARL algorithms. The method focuses on disrupting the underlying coordination structures between agents by partitioning agents into groups and using mutual information to quantify cross-group influence. Based on this, the authors carefully design attacks that selectively mask observations and perturb actions for one of the two groups. In doing so, the authors formalize their framework as a Joint-Adversarial Dec-POMDP and show empirically that IBAL improves robustness against both targeted adversarial attacks and non-parametric perturbations on the widely used SMAC benchmark.

**Compliance With Llm Reviewing Policy:**

Affirmed.

**Final Justification:**

The authors have successfully addressed my main concerns. I increased the score from 4 to 5.

**Key Questions For Authors:**

- How scalable is the proposed method to larger games (e.g., > 10 players)? Is it computationally efficient in such settings as well?
- Is hyperparameter tuning a problem for this method? In practice, is there a rule of thumb for tuning the newly introduced hyperparameters?
- Have the authors tried something more clever than just randomly partitioning the two groups?
- Is the proposed method applicable to real-world observation features (e.g, RGB-images)? How do the authors propose to filter out specific semantic dimensions?

**Strengths And Weaknesses:**

Strengths:
- The authors introduce a novel model for adversarial training MARL under CTDE. The model is well-motivated and justified experimentally.
- Strong empirical results on SMAC: The authors compare their method against state-of-the-art techniques, with the proposed method demonstrating promising results in challengings maps.
- The ablation studies work well, justifying each component of the method.
- The proposed method is not computationally heavy, compared with other state-of-the-art baselines.
- Overall, the paper reads well and the authors have made a great effort to explain their method both technically and intuitively.

Weaknesses:
- Although SMAC is arguably a strong benchmark for MARL, it is highly recommended that the authors evaluate their method on other MARL benchmarks as well (e.g., LBF and RWARE for fully cooperative MARL, or even GRF). So the main concern here is that it is likely that the proposed method has been designed exclusively for tasks like SMAC.

---

> ### Author Rebuttal · Authors · 2026-03-31
>
> ## Reviewer wy4t
>
> We sincerely thank the reviewer for the thoughtful and in-depth questions. We would like to address the reviewer’s concerns as follows. We will incorporate all of the points addressed in our response into the revised manuscript.
>
> **Weakness 1 (Additional benchmarks)**. We appreciate the reviewer’s valuable suggestion. To further validate IBAL beyond SMAC, we conducted additional experiments on two environments suggested by the reviewer: **Level-Based Foraging (LBF)** and **SMACv2**, the latter also suggested by Reviewer 8L32. For LBF, we evaluated Foraging-20x20-np-6f-s3 with $n = 5$ and $10$ agents. For SMACv2, we considered three scenarios: terran_5_v_5, zerg_5_v_5, and protoss_5_v_5. Table R1 reports the success/win rate. Across both environments, IBAL consistently achieves the highest return in all settings. While baseline performance degrades substantially under interaction-breaking attacks, IBAL remains strong against most attack types and also shows improved natural performance in highly stochastic environments such as SMACv2. These results indicate that **IBAL generalizes well to diverse MARL environments beyond SMAC.**
>
> **Table R1.** Additional Comparison on LBF and SMACv2 (Success/Win rate).
>
> |Attack|Model|LBF($n=5$)|LBF($n=10$)|terran_5_v_5|zerg_5_v_5|protoss_5_v_5|
> |---|---|---|---|---|---|---|
> |Nat.|QMIX|80.0±1.9|**96.5±2.1**|60.9±2.2|45.3±11.0|54.7±20.8|
> ||Rand.|80.2±7.8|93.7±0.8|68.0±5.5|42.2±2.2|60.8±6.8|
> ||FGSM|76.6±6.9|92.2±1.8|60.9±2.2|40.6±4.5|46.9±17.7|
> ||ATLA|73.0±3.4|93.5±0.7|65.9±8.4|42.2±2.2|59.4±13.3|
> ||IBAL(ours)|**82.7±1.9**|96.0±1.4|**78.1±4.4**|**51.6±11.1**|**77.2±2.2**|
> |Rand.|QMIX|73.5±1.4|92.1±3.0|55.6±0.9|24.7±4.9|37.5±8.9|
> ||Rand.|77.9±0.8|93.0±1.1|64.2±4.3|44.5±9.9|68.0±7.7|
> ||FGSM|72.6±3.7|94.0±4.2|60.9±15.5|42.8±1.4|48.4±6.6|
> ||ATLA|72.0±2.6|94.0±1.4|57.8±11.0|36.5±1.4|51.6±2.2|
> ||IBAL(ours)|**81.8±3.3**|**97.1±0.5**|**72.2±4.9**|**46.9±4.4**|**75.0±4.4**|
> |FGSM|QMIX|68.5±6.4|88.8±4.9|39.1±2.2|21.9±4.4|39.1±15.5|
> ||Rand.|72.2±4.5|90.1±3.3|49.2±5.5|27.3±3.3|40.6±13.3|
> ||FGSM|76.8±3.0|94.0±0.3|57.8±2.2|40.6±17.8|**70.3±2.2**|
> ||ATLA|72.0±7.1|92.9±0.1|56.3±8.8|26.6±6.6|41.9±2.7|
> ||IBAL(ours)|**78.4±2.0**|**95.0±0.5**|**64.7±3.1**|**42.2±15.5**|61.9±8.0|
> |Ours|QMIX|47.5±5.0|56.1±2.3|10.9±6.6|11.3±1.8|5.4±1.2|
> ||Rand.|46.5±3.5|59.1±2.2|19.5±3.3|17.1±6.7|20.2±2.4|
> ||FGSM|56.5±3.5|76.4±1.0|17.2±11.0|26.5±2.1|7.8±2.2|
> ||ATLA|55.0±1.4|50.2±1.7|20.9±1.3|16.5±2.1|4.6±2.3|
> ||IBAL(ours)|**70.1±2.1**|**94.7±0.4**|**48.8±2.5**|**33.1±2.8**|**50.4±2.0**|
>
> **Question 1 (Scalability).** As analyzed in Appendix F.4, the additional cost of IBAL mainly comes from MI estimation and grows with the number of agents. Empirically, the training time increases from about 50 hours on 2s3z (5 agents) to 65 hours on MMM (10 agents), which is about a 30% increase. **This overhead is comparable to that of other MARL methods** and does not constitute a disproportionately large share of the overall training cost. For example, QMIX shows a 37% increase under the same comparison. These results indicate that IBAL scales reasonably with the number of agents.
>
> **Question 2 (Hyperparameter tuning).** IBAL involves two key hyperparameters, the maximum group size $K$ and the masking budget $L$. Both are designed to scale proportionally with the number of agents so that the attack is neither excessively disruptive nor too weak. Although the method shows some sensitivity to these hyperparameters, the number of agents is known in advance for each environment. Therefore, **we expect that setting $K$ and $L$ in proportion to the agent count, as suggested in our paper, provides a practical guideline** for achieving strong performance across environments.
>
> **Question 3 (Advanced grouping strategies).** The reviewer raises an important point. In our current implementation, we use random group partitioning to expose the model to diverse group configurations. As suggested, more advanced strategies could also be considered. For example, one direction is to estimate mutual information between agents and prioritize groups with stronger inter-agent dependency. Another, also suggested by Reviewer 8L32, is to model agent relationships as an interaction graph and identify the groups whose disruption would most severely impair cooperation. We did not explore these extensions in this work due to the existing complexity of our method, but we agree that they are valuable directions for future research.
>
> **Question 4 (Extension to image observations).** While IBAL is designed for vector-based observations in this work, it can be naturally extended to image-based settings. For example, the input image can be partitioned into a 2D grid, and at each time step, masking can be applied to the grid region with the highest mutual information. More broadly, the core mechanism of IBAL is not tied to a specific observation modality, and we believe it can be extended to image-based observations in a straightforward manner.

---

> > ### Author Rebuttal · Reviewer_wy4t · 2026-04-02
> >
> > I would like to thank the authors for their response. I will increase my score.

---

> > > ### Author Response · Authors · 2026-04-05
> > >
> > > We sincerely thank the reviewer for the thoughtful feedback and for the improved evaluation of our work. The additional benchmark experiments and questions suggested by the reviewer were extremely valuable for the advancement of our paper. We will incorporate the responses discussed in the review into the revised manuscript. We thank the reviewer again for their valuable comments.

---

### Decision · Program_Chairs · 2026-04-30

**Decision:**

Accept (regular)

**Comment:**

Summary: The submitted work introduces Interaction-Breaking Adversarial Learning (IBAL), a framework designed to make multi-agent reinforcement learning more robust against attacks that target agent communication and coordination. By using an information-theoretic approach, the authors develop adversarial perturbations that disrupt interaction structures, training agents to maintain performance even when coordination is compromised.

Decision: The reviewers praised the work for its strong empirical results across various attack scenarios and for addressing a less-explored area of MARL robustness. An unaddressed weakness is the lack of a formal theoretical guarantee regarding the convergence or optimality of the training objective under these specific interaction-breaking constraints. Despite this weakness, there is enough enthusiasm for the paper to be accepted, as reviewers noted that the empirical gains over existing baselines are quite significant. The AC recommends weak acceptance.